

# Droughts of the Early 19th Century (1790-1830) in
# Northeast Iberian Peninsula: Integration of historical
# and instrumental data for high-resolution
# reconstructions of extreme events
Josep Barriendos[1], María Hernández[2], Salvador Gil-Guirado[3],
Jorge Olcina Cantos[2], Mariano Barriendos[4]
1: CREAF, Autonomous University of Barcelona, Barcelona, Spain; (j.barriendos@creaf.uab.cat).
2: Department of Regional Geographical Analysis and Physical Geography, University of Alicante,
Alicante, Spain; (maria.hernandez@ua.es); (jorge.olcina@ua.es).
3: Department of Geography, University of Murcia, Murcia, Spain; (salvador.gil1@um.es).
4: IDAEA-CSIC, Institute of Environmental Assessment and Water Research, Spanish Research Council,
Barcelona, Spain (mariano.barriendos@idaea.csic.es).
*Correspondence to*: Josep Barriendos (j.barriendos@creaf.uab.cat)
**ABSTRACT.**
Drought represents a prevalent climate risk in the Mediterranean region. In the context of climate
change, an increase in both frequency and intensity is anticipated over the next century. In order to
effectively manage future scenarios where global warming overlays natural climate variability, a thorough
analysis of the nature of droughts prior to the industrial age is imperative. This approach incorporates an
extended temporal scale into the study of severe droughts, enabling the identification of low-frequency
drought events that occurred before the instrumental period. The objective of this study is to examine the
occurrence and magnitude of extreme droughts lasting over a year in the Spanish Mediterranean Basin
during the Early 19th Century (1790-1830). To achieve this objective, the research integrates the use of
instrumental observations and information derived from historical documentary sources with daily to
monthly resolutions (e.g. rogation ceremonies). The findings reveal that drought episodes were more
frequent and severe during the Early 19th Century than in the second half of this century. Moreover,
drought episodes of similar severity were rare throughout the 20th Century. Only in the current context of
climate change, over the last two decades, has a pattern of high drought severity been identified that
resembles the severity found during the Early 19th Century (especially between 1812 and 1825). This
study underscores the presence of high variability in drought patterns over the last centuries, justifying the
need for intensified research on drought episodes with high temporal resolution for extended periods.
**KEYWORDS.**
Early 19th Century, Documentary Sources, Droughts, Drought Indices, Meteorological records, Spanish
Mediterranean Basin.





## 1. INTRODUCTION

Drought is a climate phenomenon defined as a prolonged absence of precipitation that can last for a few weeks to periods of up to several years (IDMP, 2022). According to the IPCC, drought is an exceptional period of water shortage for existing ecosystems and the human population (due to low rainfall, high temperature and/or wind) (IPCC, 2022). Despite their complexity as a natural phenomenon, droughts should not be confused with aridity, desertification or other related natural risks such as forest fires or heatwaves (IDMP, 2022; Van Loon, 2015). Drought, as a prolonged lack of precipitation, can be classified depending on the impacts on the environment and society resulting in distinct types of droughts such as meteorological, hydrological, agricultural and social (Wilhite & Glantz, 1985).

Meteorological drought is defined as a prolonged period with abnormal rainfall deficit for a large region and for a long period of time (Mishra & Singh, 2010; IPCC, 2022). This absence of rain is transmitted to the hydrological system by affecting soil moisture and groundwater input, ultimately reducing surface water levels (Van Loon & Van Lanen, 2012). Thus, hydrological drought is defined as a period with large runoff and water deficits in rivers, lakes and reservoirs (Nalbantis, 2008; IPCC, 2022). It also has effects on groundwater and surface hydrology (Mishra & Singh, 2010; Wilhite & Glantz, 1985). This deficit causes a reduction in water supply to plant roots leading to agricultural and ecological drought (Van Loon, 2015). This type of drought consists of a period of abnormal soil moisture deficit, which results from combined shortage of rainfall and excess evapotranspiration (Sivakumar, 2011a; IPCC, 2022). Consequently, during the growing season, it impinges on crop production, leading to reduced yields and even crop failure (Mishra & Singh, 2010; IPCC, 2022). The different impacts of drought mentioned above, such as the reduction in water levels or crop failures, have a direct effect on human societies (Van Loon, 2015). These effects on society caused by a prolonged drought over time are defined as social drought (Van Loon, 2015). The main impact of drought on society consists on shortages or limitations in the availability of the water resource and the failure of water supply for different uses: the worsening of agricultural production, the decrease in energy or industrial production, problems in the supply of drinking water, or limitations in any recreational or ornamental use of water (Eslamian, et al., 2017; Mishra & Singh, 2010).

All drought types can be characterised by several items: (1) severity (i.e., expressed through the rainfall values themselves as well as through ceremony levels of *pro pluvia* rogations), (2) duration (i.e., from onset to end), (3) spatial extent (i.e. area of impact) and (4) frequency of occurrence (Nalbantis, 2008). Long droughts can cause serious hydrological imbalance gradually increasing its severity (Wilhite & Glantz, 1985). While magnitude, duration and recurrence are necessary drought features to assess the physical impacts of droughts on a territory, the vulnerability of the society in relation to degree of exposure and strategies to cope with the physical hazard are fundamental for a comprehensive evaluation of climate risk. Beyond contributing to direct water scarcity, droughts affect agriculture, hinder the production of energy, the access to fresh water and may aggravate political tensions connected to water rights (Gorostiza et al., 2021). Moreover, the socio-economic impacts of drought generally persist even when the episode of meteorological or hydrological drought ends and the volume of rainfall returns with regularity (IDMP, 2022).



Despite the importance of droughts and their capacity to seriously affect the economic and
productive activities of societies, the level of knowledge on this natural phenomenon contrasts with that
of other natural hazards (Van Loon & Van Lanen, 2012). For these reasons it is justified to conduct a
more detailed and systematic study of drought events. It will also take into account the analysis of
specific episodes of lower frequency and greater severity, which may provide additional information on
long term drought behaviour (Olcina, 2001a). Of particular interest are those that have occurred within
the framework of the Mediterranean, where drought is an intrinsic phenomenon of the climate of the
region (Olcina, 2001b).
In general terms the Mediterranean climate in the Iberian Peninsula is characterised by a highly
irregular rainfall, both inter-annual and intra-annual (Martín-Vide & Olcina, 2001). Another characteristic
is the pronounced aridity during the warm season (summer) (WMO, 2023). Additionally, presents
important variations in the intra-annual distribution of precipitation depending on the region (Martin-
Vide, 1985). On the eastern of Iberian Peninsula and Balearic Islands, Mediterranean climate type has
two main varieties in relation to the seasonal distribution of rainfall. Between the provinces of Girona and
Almería, main rainfalls are in autumn, with a secondary peak in spring. In the southernmost provinces
(coast of Granada, Málaga and Cádiz) the most abundant rains are recorded in the autumn and winter
months (AEMET, 2011). Autumn rains are mainly linked to cold drop atmospheric situations (Sánchez-
Almodóvar et al., 2022). In the spring, along with this cold drop configuration, convective rains (storms)
take on a prominent role. In the winter, Atlantic storms with a southern trajectory (Gibraltar strait) are
frequent in the south of the Mediterranean strip. In either of these two varieties, summer is always the
season with the lowest rainfall contribution. Annual quantities decrease from north to south between
Girona and Almería (Serrano-Notivoli et al., 2018). In the southern section of this region, rainfall
increases again due to the greater contribution of winter rains (Mathbout et al., 2020). For this reason,
spring and autumn are the key seasons that balance the annual water input. Strong droughts occur when
the summer and winter lack of precipitations connect due to an extraordinary lack of rainfall on the rainy
seasons. These seasonal aspects determine a high temporal variability of water reserves on the Northern
Spanish Mediterranean Basin (Kim & Raible, 2021). In addition, this high vulnerability is magnified by
the eventual impacts caused by droughts (González-Hidalgo, et al., 2018). In this regard, together with
drought, water management in the Mediterranean region have always been a challenge, but now it is
exacerbated within the context of climate change (Hohenthal & Minoia, 2017). Additionally, serious
problems have derived from greater water demands in result of population increase and the spread of a
lifestyle model based on mass consumption of goods and services (IDMP, 2022). In this respect, the
Mediterranean region is a clear example of imbalance between water demand and water availability. As a
result, in recent decades it has become one of hotspots areas impacted by climate change. Along with the
already detected temperature increase, since the beginning of the 21st Century there is also the added
challenge of increased rainfall variability (Barrera-Escoda & Cunillera, 2011). In this context, along with
the increased rainfall irregularity, extended dry periods occur with greater frequency and severity
(Marcos-García, et al., 2017; Kim & Raible, 2021). Therefore, drought in the Spanish Mediterranean
Basin is one of the natural risks with the greatest impact, due to its capacity to cause simultaneous effects



on different levels: environmental, economic, social, etc., and also its capacity to last for a long time
(Walker et al., 2010).

Because of the impacts of extreme hydrometeorological phenomena in the Mediterranean, such
as droughts and floods, observation of their behaviour in the recent past is justified. Previous work on the
reconstruction of rainfall on a long-time scale generally shows situations of rainfall shortage (Pauling, et
al., 2006; Camuffo, et al., 2012; Smerdon, et al., 2017). As well the results obtained from instrumental
data series highlight that rain shortage (droughts) are perceived at the seasonal level. The present work
aims to analyse the most extreme phenomena detected in the aforementioned paper, but using data that
allow us to analyse rainfall deficits at the monthly level. Unfortunately, for the Iberian Peninsula as a
whole in the study under review there is only one instrumental data series. For this reason, the
instrumental data from the Barcelona station (1786-2022) complement the historical data from the
MILLENIUM project ("European climate of the last millennium", Code: IP 017008-2). The combination
of instrumental and historical data has been used to study specific periods of anomalous temperature and
rainfall conditions. One such widely studied case is the anomaly of the non-summer year of 1816, a
consequence of the Tambora volcanic eruption (Trigo, et al., 2009; Luterbacher & Pfister, 2015).

The Spanish Mediterranean basins are currently experiencing a situation of severe rainfall
shortage. Due to this serious situation, it is necessary to find references of droughts of equal or greater
magnitude in order to understand the characteristics of these phenomena in their most extreme behaviour.
Studies carried out on the Iberian Peninsula to study historical droughts using historical data show
significant results obtained from the use of rogations as a data (Dominguez-Castro, 2012; Tejedor, 2019).
These studies make it possible to identify the importance of the 19th century for its study, highlighting
specific years such as 1817 or 1824 (Dominguez-Castro, 2012). Despite these results, the data used in
these studies were applied to yearly resolutions. The need for knowledge of past droughts adds to the
need to expand the detail of existing studies on historical droughts in the study area.

Historical data allow us to observe the behaviour of droughts in much more distant historical
periods than those of the instrumental precipitation data series. Therefore, this data would allow us to
improve the knowledge of drought natural variability over a long-time scale than the instrumental period.
Also, this longer timescale would help to study drought return periods on centennial scale (i.e. lower
frequency droughts) and the duration and magnitude of past extreme droughts (note that only a handful
are available during the instrumental period).  In the case of droughts, it is crucial to know those episodes
which occurred in the past and whose severity, extent and duration were exceptional (Gil-Guirado, et al.,
2016).

According to all the reasons exposed above, in the current paper we will discuss the topic of the
extreme droughts that affected the Mediterranean Basins of the Iberian Peninsula during the Early 19th
Century (1790-1830). The detailed study of drought events during this period is justified by the physical
and social reasons that underline their exceptionality. The severity of the different droughts recorded,
their cumulative duration and the impact they had on the societies of the Spanish Mediterranean Basins
do not have an equal magnitude in the recent collective memory. On the other hand, this period has been
studied relatively well, thanks to climate reconstructions for the beginning of the 19th Century based on



natural and historical *proxy* data and the first instrumental meteorological data series (Brönnimann et al.,
2018b; Prohom et al., 2016).
This paper focuses on the impacts caused by meteorological droughts because of the nature of
the data used. The main sources of information used for the analysis of droughts in the historical period
are the instrumental precipitation data sets of Barcelona (Catalonia, NE Spain) and the historical data of
rogations (Spain, with higher density for Catalonia). The case of the rogations differs from that of the
instrumental series, since the former focuses on the lack of precipitation while the rogations would allow
the analysis of agricultural drought (Brázdil et al., 2018). However, rogations also allow meteorological
monitoring of the natural phenomenon, as the ceremonies itself are interrupted when an improvement in
rainfall is detected. This is because of the daily level of detail of the rogation system as a source of
information (Martín-Vide & Barriendos, 1995). The very etymology of the rogations (*pro pluvia*, to
obtain rain as usual) demonstrates the meteorological nature of the ceremony. Their purpose was not
directly to obtain a large harvest, but to achieve a good rainfall episode.
### 1.1.    Research background
The Early 19th Century (1790-1830) occurred during the climate episode named as the Little Ice
Age (hereafter, LIA) between the fourteenth and nineteenth centuries (Grove, 1988). This climate
oscillation was clearly characterised by lower average temperatures with respect to the previous episode
(Medieval Warm Period) and the subsequent episode (Current Global Warming) (Fischer et al., 2007).
Another significant aspect of the LIA is the irregular behaviour of rainfall, with a clear increase in the
frequency and magnitude of severe hydrometeorological events (Barriendos et al., 2019, Oliva et al.,
2018). In the case of the Iberian Peninsula, different oscillations were observed including increases in
heavy rains or droughts throughout this period (Barriendos, 1996). One of the most exceptional
oscillations is called Maldà Oscillation, which occurred between 1760 and 1800 (Barriendos & Llasat,
2003). The Maldà Oscillation was characterised by simultaneous increases in the frequency of heavy rain
events, alternating with droughts. The alternation of extreme rainfall and droughts events had strong
social and economic impact on the Iberian Peninsula. Specifically, the sequence of droughts, cold snaps
and snowfalls had serious direct consequences on agriculture, while consecutive floods also damaged or
destroyed many infrastructures. Furthermore, during the period of the Maldà Oscillation there was an
emergence of uncommon epidemic diseases, such as smallpox or yellow fever viruses, occurring at the
same time than more common diseases such as epidemic malaria or typhoid (Barriendos & Llasat, 2003;
Alberola, 2010; Alberola & Arrioja, 2018).
Within the LIA, the Early 19th Century was characterised by an abnormally low amount of
emitted solar radiation, which generated an overall decrease in the amount of solar radiation arriving to
the Earth (Prohom et al., 2016). In addition to this external forcing factor, climate variability at the end of
the LIA was also affected by several volcanic eruptions that occurred between 1790 and 1830 (Fang et al.,
2023). Of these eruptions, 247 had an VEI ≥ 2; 35 had a VEI ≥ 3; 16 had a VEI ≥ 4; 2 had a VEI ≥ 5; 1
had a VEI ≥ 6 and 1 had a VEI ≥ 7 (Global Volcanism Program, 2023). Among these volcanic eruptions,
stand out a sequence of large explosive volcanic eruptions (Wagner & Zorita, 2005; Prohom, 2003):
*Unknown* (1808), Tambora (1815), Galunggung (1822) and Cosigüina (1835). Some studies indicate that



the high intensity volcanic eruptions, occurring between the LIA and the current Global Warming, led to
a decrease in temperatures, together with an increase in rainfall irregularity in the study area (Gil-Guirado
et al., 2020).
Among the three eruptions of the Early 19th Century, the 1815 Tambora eruption is considered
one of the most significant of the past two thousand years in terms of the particles emitted (Raible, et al.,
2016). Also, it is considered as the cause of the most pronounced climate anomaly of the first third of the
19th Century (Brönnimann et al., 2018b). Due to his outstanding volcanic explosivity (VEI 7), this
eruption was the largest and most devastating eruption recorded in the historical age and is considered to
be responsible for the "year without a summer" of 1816 reported across Europe and North America
(Trigo et al., 2009; Luterbacher & Pfister, 2015). Central Europe, Western Europe and Northern Europe,
with temperatures recorded of between 2 to 3ºC below the average in areas of Spain and Portugal (Pfister
& White, 2018) were the greatly affected regions by this "year without a summer". During that summer
the number of rainy days almost doubled and cloudy days were more frequent in the whole of Europe and
North America. Alterations in the usual general atmospheric circulation pattern and its centres of action
were also reported as a result of cooling due to the direct effect of the reflection of incident radiation
associated to the presence of volcanic aerosols (Brönnimann et al., 2018b).
This study has found a time period in which there is an accumulation of particularly severe drought
episodes. This period coincides chronologically with Dalton Solar Minimum and an anomaly in volcanic
activity (eruptions of Tambora and other volcanoes mentioned). Obviously, the chronological coincidence
does not presuppose any cause-effect relationship between the anomalies in solar and volcanic activity
and the pluviometric anomalies under study.
**1.2.    Historical Droughts Studies in Spain**
The analysis of historical droughts in Spain dates back to studies by Manuel Rico y Sinobas in
the mid-19th Century, in which he analysed the impacts of drought episodes on agriculture. His main
objective was to compile records in order to obtain a broad temporal dimension of the phenomenon (Rico
y Sinobas, 1851). Subsequently, and until the beginning of the 1990s, only sporadic studies were carried
out that were in some way related to events (Bentabol, 1900). One exception is the study by Couchoud
(1965), who analysed the region of Murcia in depth (SE Spain) based on a detailed compilation and
analytical process. In 1994, two PhD theses on historical climatology that engaged with droughts were
defended in Spain (Barriendos, 1994; Sánchez Rodrigo, 1994). They constitute benchmark studies in the
research on this topic. From this decade onwards, there has been a proliferation of studies and
publications in which drought is taken into consideration (see, among other, Sánchez Rodrigo, et al.,
1994; Martin Vide & Barriendos, 1995; Sánchez Rodrigo, et al., 1995; Barriendos, 1997; Barriendos &
Martin Vide, 1998; Sánchez Rodrigo, et al., 1998), including manuals on natural risks (Olcina, 2001a).
More recently, a new PhD thesis (Gil-Guirado, 2013) once again insisted on the need to study historical
droughts in the Spanish Mediterranean Basin based on a quantitative approach.
In addition to PhD theses, there are also recent publications focused on the study of historical
droughts using a quantitative approach. An example that actually corresponds to the period analysed in
present work is the paper focused on droughts for the Iberian Peninsula (1750-1850) (Dominguez-Castro



et al., 2012). This article approaches the severe episodes of historical droughts by means of rogations at
annual resolution. Other studies have continued this line of research in the Iberian Peninsula (Trigo et al.,
2009; Fragoso et al., 2018; Tejedor et al., 2019; Bravo-Paredes, 2020) and even in more detail for the
Ebro basin (Cuadrat et al., 2022). The availability of pro-pluvia rogations in the Hispanic Monarchy
extended beyond the Iberian Peninsula, as evidenced by works in Mexico and all Central American
countries (Garza-Merodio, 2017; Alberola & Arrioja, 2018; Ramírez-Vega, 2021). Rogations are a
liturgical mechanism used in other Catholic countries and therefore these studies can be extended to this
broader religious sphere (Garnier, 2019; Pfister, 2018). Finally, the amount of information that is
becoming available is already being organised in comprehensive databases such as AMARNA or in
international initiatives (Domínguez-Castro et al., 2021).
Parallel to the research based on rogations, the study of historical droughts in the Iberian Peninsula has
also been carried out through the analysis of ancient instrumental precipitation data series as well
(Prohom et al., 2016). Or with the combination of data on rogations and precipitation series analysed by
means of drought indices (Tejedor et al., 2019). These studies allow us to observe severe droughts based
on inter-annual variability.

## 1.3.    Objectives

The main objective of this study is to analyse the patterns of drought episodes that affected the
Northeast of the Iberian Peninsula during the Early 19th Century (1790-1830) using instrumental and
historical sources. This period that corresponds to the last stages of the Little Ice Age was chosen due to
severity of drought occurring in the Mediterranean Basins of the Iberian Peninsula. Additional objectives
of this study are: 1) to qualitatively and quantitatively extend the AMARNA database on climate risks
(*Arxius Multidisciplinars per a l'Anàlisi del Risc Natural i Antròpic*, from catalan: Multidisciplinary
Archives for the Analysis of Natural and Anthropogenic Risk) to incorporate droughts and different social
processes linked to environmental impact in addition to hydro-meteorological excesses (Tuset et al.,
2022); 2) to compile and describe the variability of extreme hydrometeorological events (heavy rainfall
and droughts) in the Spanish Mediterranean Basin during the Early 19th Century. In order to study how
the opposite extreme events behave and interact with each other. Also to understand if the behaviour of
past hydrometeorological extremes is similar to the modelled behaviour for the future in the study area. In
addition, the spatio-temporally coherent periods of climatic anomalies have among their main
characteristics the increase in rainfall irregularity in the study area (Gil-Guirado et al., 2016); 3) to
characterise the drought episodes, analysed from historical data, considering their duration, extension and
severity in high resolution for the period analysed; and 4) to analyse the entire instrumental precipitation
data series of Barcelona (1786-2022) for the whole duration of the series in order to characterize periods
of drought.
In order to fulfil these objectives, the paper analyses the historical and instrumental data
available in the Spanish Mediterranean Basins, using different time and spatial scales. The socio-
environmental context during the Early 19th Century is analysed using data compiled from historical
documentary sources, namely the records of the *pro pluvia* rogation ceremonies held in the main villages
of the affected regions. These data are compared with the analysis of the instrumental precipitation data





series of Barcelona (1786-2022) based on different statistical techniques, including the use of three
drought indexes: SPI (McKee et al., 1993), SPEI (Vicente-Serrano et al., 2010) and Deciles (Gibbs &
Maher, 1967).
The article focusses on analysing climate variability during the Early 19th Century period and
provide the state of the art on droughts in historical perspective in Spain and Europe as a whole.
Subsequently, the results obtained are presented through graphic and cartographic resources.
**2. MATERIALS AND METHODS**
**2.1.    Sources of information**
The sources of information used to analyse droughts in the Early 19th Century consist mainly of historical
data and the Barcelona instrumental precipitation data series ranging from 1786 to the present day. The
historical data on droughts in the Spanish Mediterranean Basin during the Early 19th Century was
obtained from Documentary sources of public administrations and ecclesiastical institutions complied in
the AMARNA database (Barriendos & Barriendos, 2021; Tuset et al., 2022). AMARNA is an archive that
compiled climate historical episodes from different documentary sources which are geo-referenced and
classified into numerical categories on a daily resolution. The information from AMARNA refers to any
type of extreme meteorological event and its social impacts. Events about which there is more
information are those relating to water excess (persistent rainfall, pluvial and fluvial floods) and rainfall
deficits (droughts). The total number of records for the period EC 1035-2022 amounts to slightly more
than 19,000 cases, organised in more than 5,500 episodes (Tuset et al., 2022). Sources of information are
mainly administrative and private documentary sources, with direct descriptions of events and their
impacts. The institutional documentary sources also provide systematic and continuous records over time
throughout the existence of the institution, with resources and conditions that favour the conservation and
access to the documents (Martín-Vide & Barriendos, 1995; Brönnimann et al., 2018a). Water deficits are
obtained from the records of *pro pluvia* rogation ceremonies (cultural-historical proxy) from municipal
and local ecclesiastical sources (Brádzil, et al., 2018). Rogations are the main *data proxy* in order to
identify and compile information on droughts in the Spanish Mediterranean Basin. The records of these
ceremonies are generated and initiated by public authenticators in collegiate administrative bodies
(municipal councils, cathedral councils), which guarantees the reliability of the document itself and the
veracity of the information contained therein. The rogation records contain reliable and homogeneous
information due to their institutional origin and the formal rigidity of the related liturgical procedures
(Brádzil, et al., 2018). The documentary record of the rogation ceremony informs of the location, the date
and duration of the drought conditions. With respect to the severity of the event, the application of a
specific methodology based on the type of liturgical acts used enables their classification by categories
and their numerical indexing (Martín-Vide & Barriendos, 1995; Barriendos, 1997). As a complement to
these administrative sources, AMARNA also uses private personal sources, such as appointment books,
memoirs or chronicles.
Rainfall excesses are also found in the same administrative documentary sources as the deficits
and their cataloguing and numerical classification procedure is also based on objective indicators. In the



1990s, simple and easy to cross reference classification criteria were proposed for all of the European
basins, based on the levels of river overflows and the damage recorded (Barriendos & Martín-Vide, 1998;
Brázdil et al., 1999). The first studies that used these information sources in the area of study sought to
conduct an overall reconstruction of the climate variability through the generation of weighted annual
indices (Barriendos, 1996; Barriendos, 2005). Subsequent studies extended the analysis with annual
indices for different locations of the Spanish Monarchy, both on historical floods (Barriendos & Sánchez
Rodrigo, 2006) and for droughts (Domínguez-Castro et al., 2008; Domínguez-Castro et al., 2012;
Sánchez Rodrigo & Barriendos, 2008, Tejedor et al., 2019; Gil-Guirado, et al., 2019).

In addition to the analysis of historical data, the second part of the study consists in the statistical
analysis of the instrumental precipitation data series of Barcelona spanning from 1786 to 2022.
Unfortunately, the Barcelona set is the only continuous rainfall series available in the study area for the
Early 19th Century. This information is scarce for such a large geographical area, but the Barcelona series
is located in the area with the most historical information available for this period. Therefore, the joint
analysis of instrumental and historical information is relatively consistent.

The Barcelona rainfall series has been compiled by consulting several documentary sources and
different sets of records. The principal source was elaborated by the Meteorological Service of Catalonia,
*Servei Meteorològic de Catalunya* (SMC) (Prohom et al., 2016). This was compiled from different
institutional observers who generated records during the 18th and 19th centuries in the centre of
Barcelona (at around 30 meters above sea level). For the 20th Century the records were generated at the
Fabra Observatory, placed outside of the city (at the Tibidabo mountain, at 412 meters above sea level).
The analysis of these sources has enabled the homogenisation of the monthly precipitation data sets from
1786 to 2014. To complete the SMC series up to the year 2021, instrumental records from a private
observatory in the Can Bruixa neighbourhood of Barcelona was used. For the year 2022, data was from
the official SMC Raval automatic station (University of Barcelona). These two sets are validated by the
SMC and their data have been collected in the centre of the city of Barcelona, making their values closer
to those collected at the beginning of the series.
**2.2.   Indexation system of historical climate data**

This study is based on the use of information on a daily scale drawn from the historical data
obtained from the AMARNA database. This information is organised into cases and episodes. Every
episode consists of a group of cases or records of different dates and locations which provide information
about the impact and duration of each episode. Cases are the basic units of documentary record in which
there is mention of some kind of impact on the water deficit. They may be decisions by the authorities to
initiate or continue pro pluvia rogations, qualitative records of rainfall within a drought episode, or
records of the decisions taken by the authorities to end the rogations once the drought is considered to be
over. The cases and episodes are classified into five categories and fifteen sub-categories (Barriendos &
Barriendos, 2021) (Table 1). These fifteen thematic subdivisions proposed correspond to the highest
degree of detail observable in the documentary and bibliographic sources consulted (Table 1). For the
specific case of drought episodes (DR), these come mostly from records of the celebration of *pro-pluvia*
rogation ceremonies. These records provide information on both the duration and severity of drought



events. By having the dates on which each ceremony is held, we can identify both the beginning and the
end of the rogations, along with increases in severity. The type of ceremonies held in order to ask for rain
will define the severity of the episode, which are organised between 1 and 5 (Martín-Vide & Barriendos,
1995; Barriendos, 1997). A level 1 rogation marks the beginning of each drought episode and a gratitude
ceremony marks the end of the episode. Each drought episode will thus have a different duration and its
severity will be defined by the ceremonies between the first rogation and the closing of the ceremonies.

| CATEGORIES | | SUB-CATEGORIES | |
|---|---|---|---|
| Code | Name | Code | Name |
| ERE | Extraordinary Rainfall Event | FF | Fluvial Flood |
| | | PF | Pluvial Flood |
| | | PR | Persistent Rainfall |
| | | SS | Sea Storm |
| | | DR | Drought |
| ECE | Extraordinary Convective Event | HE | Hail Event |
| | | ES | Electric Storm |
| | | WS | Wind Storm |
| ETE | Extraordinary Thermic Event | CW | Cold Wave |
| | | US | Unusual snowfall |
| | | HW | Heat Wave |
| SIE | Social Impact Event | EE | Epidemic Event |
| | | PE | Plague Event |
| | | FS | Food Shortage |
| ERR | Technical mistake | ERR | Spurious case |

**Table 1: Classification system of the AMARNA database** (Barriendos & Barriendos, 2021)**.**
The AMARNA database originally only provided data on water excesses recorded in historical
periods for the Spanish Mediterranean basins (Tuset et al., 2022). An effort is currently being made to add
data on droughts to the AMARNA database. In this regard, the period from 1790 to 1830 has been a test
to see how the recently gathered data on droughts and the existing data on excess water fit together. Thus,
the work proposed in this article supposed progressing from 0 cases and episodes of drought for the Early
19th Century, to the values with which the study has been carried out. The AMARNA database is still
under development for other historical periods and therefore not yet available for public access.
The georeferencing of all the historical data compiled in the AMARNA database allowed the use
of SIG tools for the cartographic representation of this historical information. The distribution of the
droughts in the Early 19th Century have been represented both on a municipal level and with the cases
grouped by hydrographic basin. These are the Spanish administrative units for managing water resources
(Figure 1) (MITECO, 2023). The organization at a municipal level allow the analysis of the time-space
distribution of the impacts caused by different drought episodes representative of the period of study. The
different efforts to compile data on the AMARNA database on water excesses and droughts have resulted
in a very characteristic distribution of data for the case of the Early 19th Century period (Figure 1). Most
of the points with information on water excesses collected in AMARNA are located in the Spanish
Mediterranean basins. On the other hand, the information on droughts covers points all over Spain, but
with a higher density in the territory of Catalonia, between the CHE and the CIC hydrographic basins, and
Murcia city (CHS) (Figure 1). This disproportion in the amount of information between the Atlantic and
Mediterranean basins is due to the effort focused on the latter, where there is more interest in the study of



hydrometeorological phenomena. Therefore, as the title of the paper indicates, the analysis of drought on
the Notheast Iberian Peninsula uses information from the Atlantic basin of the Iberian Peninsula only as a
reinforcement or complement for a better characterisation of the episodes identified.

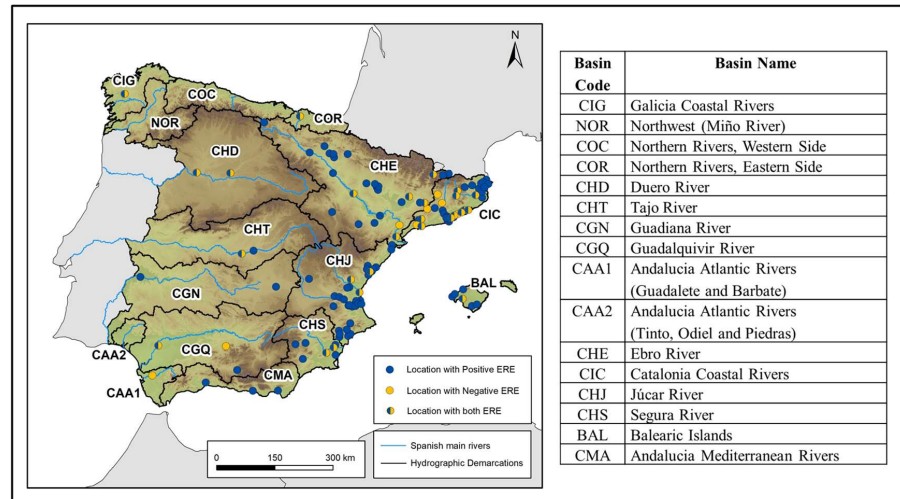

**Figure 1: Spanish hydrographic basins analysed in this study. Locations with historical information for the Early 19th Century. This specifies the locations that have records on positive ERE (FF, PF, PR and SS), negative ERE (DR) or both types of ERE.**

### 2.3. Generation of drought indices

Several drought indices were generated using the Barcelona precipitation data series (1786-
2022). In all cases, the indexes were calculated based on monthly values and for groups of 12 months.
The SPI (*Standardized Precipitation Index*) (McKee et al., 1993) was the first index calculated, which is
widely used for classifying droughts (WMO & GWP, 2016). This index enables the analysis of the
duration and variability of droughts, as well as of the wet periods and is generated based on the
transformation of the temporal precipitation data series in a standardised normal distribution (Lloyd-
Hughes & Saunders, 2002; Gil-Guirado & Pérez-Morales, 2019; Zargar et al., 2011). The second index is
the SPEI (*Standardized Precipitation Evapotranspiration Index*) (Vicente-Serrano et al., 2010), which is
similar to the SPI index, but also uses the average monthly temperature variable (WMO & GWP, 2016).
It is a relatively versatile index, simple to apply and enables analyses to be carried out for any climate
regime (Stagge et al., 2015). The third index used is the Deciles index (Gibbs & Maher, 1967), which
stands out for its applicability and simplicity, due to the facility of the calculations that it requires and the
fact that it only requires precipitation data (Hayes & Cavalcanti, 2005; Tsakiris, et al., 2007). This method
is obtained by dividing the distribution of the monthly precipitation data into deciles (WMO & GWP,
2016), which define thresholds for different water deficit conditions (Eslamian et al., 2017; Zargar et al.,

2011).

Statistical analyses were conducted with the results of the three indices. Testing of trends was
carried out using the Mann Kendall test and with the Sen slope. Analysis of breakpoints of the monthly
series was conducted using the Pettitt Test (Gil-Guirado & Pérez-Morales, 2019).



Based on the results obtained in the different drought indices, a criterion has been defined to
organise the different drought episodes detected for the Early 19th Century. This criterion takes the values
obtained with the SPI as a reference to define each episode. Each drought episode must have at least six
consecutive months with SPI values lower than "-1". The count of the total number of months of the
drought episode starts and ends when the SPI values are below "-0.75".

## 3. RESULTS

### 3.1. The hydro-meteorological extremes in Spain (1790 - 1830)

The AMARNA database used in this paper provides a total of 19115 cases spread over 5,551
episodes for the period from 1035 to 2022. For the Early 19th Century (1790-1830), the AMARNA
database provides for the whole of the Iberian Peninsula 2047 cases, which are grouped into 708 episodes
(Barriendos et al., 2019).
From the 2047 total number of cases 1789 cases correspond to ERE events (Extraordinary Rainfall
Event). Within the ERE cases, there is a clear predominance of the subcategory DR (Drought), with 64%
of the ERE cases (Table 2).

| Subcategories | Number of cases | Percentage |
|---|---|---|
| Fluvial Flood (FF) | 431 | 24.09% |
| Pluvial Flood (PF) | 40 | 2.24% |
| Persistent Rainfall (PR) | 164 | 9.17% |
| Sea Storm (SS) | 22 | 1.23% |
| Drought (DR) | 1132 | 63.28% |
| Total | 1789 | |

**Table 2: Total number of cases of the five groups making up the ERE category (Extreme Rainfall**
**Event). Elaboration from AMARNA database.**
The temporal distribution of the ERE episodes throughout the Early 19th Century reveals a
predominance of droughts with respect to the other types of ERE, but with a non-homogeneous
distribution (Figure 2). For instance, between 1790 and 1800 rainfall was abundant, so floods were more
significant than droughts in years such as 1793, 1797 or 1801 (Figure 2). This decade also stands out due
to its clear irregularity across different years, which can be related to the final part of an abnormal climate
period detected between 1760 and 1800, known as the Maldà Oscillation (Barriendos & Llasat, 2003).
The 5-years moving averages show the most pronounced episodes of droughts and water excesses during
this period. Figure 2 highlights its temporal distribution: in the first decade, positive extreme peaks were
interrupted with the drought of 1798. On the other hand, from the episode of 1807, droughts became
predominant, being particularly severe between 1812 and 1825 (Figure 2). The positive EREs cases
diminished from 1806 definitively for the rest of the Early 19th Century, while the negative EREs
increased from 1812. Between these two well defined periods exists a transition period with low number
of heavy rainfalls or droughts.



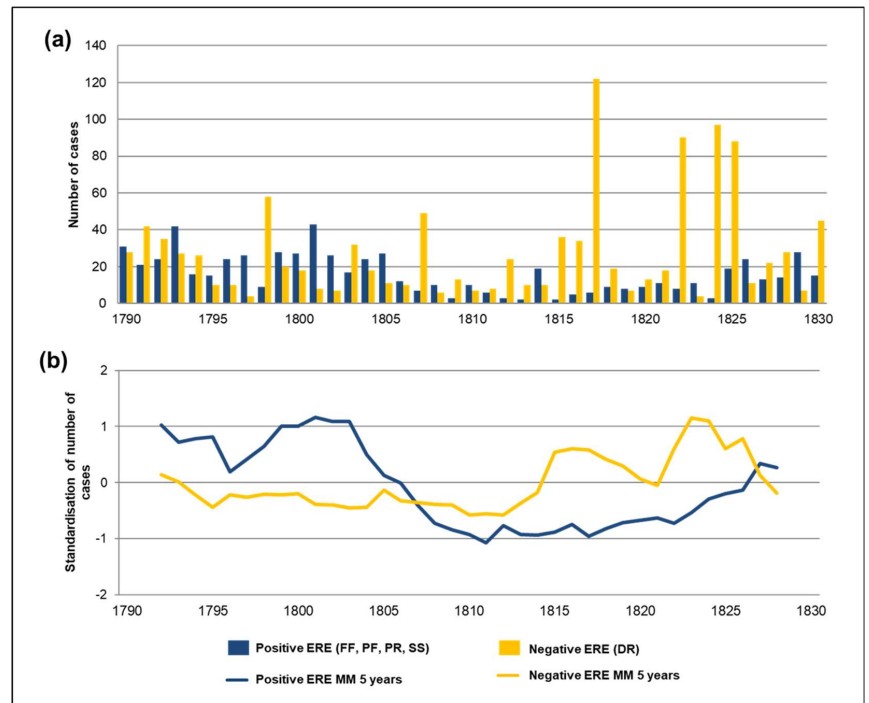

**Figure 2: a) Temporal distribution of positive EREs and negative EREs during the Early 19th Century (1790-1830). b) Annual cases of positive EREs and negative EREs during the Early 19th Century (1790-1830). 5-years moving averages of the standardised values for the positive EREs and negative EREs. Elaboration through AMARNA database.**

The geographical distribution of ERE cases for this period also provides interesting information. It is highlighted the large number of cases recorded in the Spanish Mediterranean Basin against those recorded in the Atlantic basins for the same period (Figure 3). The Guadalquivir basin (CGQ) is the only Atlantic basin with an important amount of ERE cases. The predominance of drought in the Spanish Mediterranean basins contrasts with the greater impact of the positive ERE episodes in the Atlantic basins. In the Mediterranean area, the Júcar basin (CHJ) stands out as there is a high incidence of positive ERE, unlike the dynamics of the other Mediterranean basins. This bias can be applied to the CHJ, NOR, CGN and CMA basins. For this reason, in the basins that suffer this bias, the majority of the information corresponds only to the episodes of positive ERE.

The towns that account for more than 50 cases of drought were all spatially distributed across the Mediterranean basins, except for Seville, located in the Atlantic watershed (Figure 4). Regarding the drought temporal distribution at the different cities, Murcia case is noteworthy by the regularity of drought episodes compared to the majority of the other cities that exhibit larger temporal variability (Olcina, 2001b). This fact is related to its geographical position in the South-east of Spain. Within this environment, "specific" drought events occur (the so-called "surestinas" south-eastern droughts) related to the lack of precipitation from the Atlantic and absence of Mediterranean rainfall events (Olcina, 2001b).





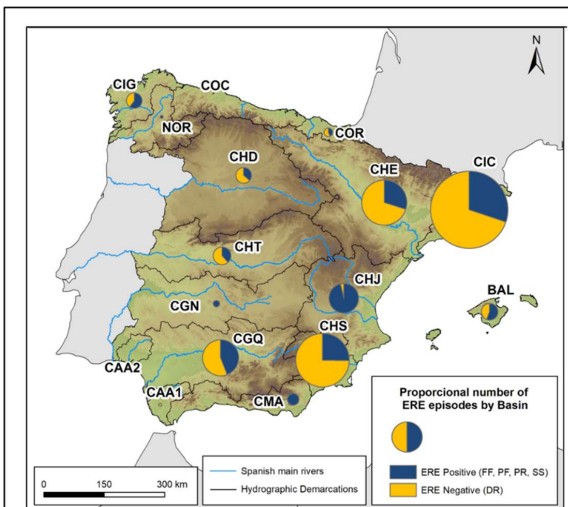

**Figure 3: Number of positive ERE cases (FF, PF, PR, SS) and negative ERE cases (DR) for the**
**different Spanish river basins during the Early 19th Century (1790-1830). A list of the full names of the basin**
**codes can be found in Figure 1. Elaboration from AMARNA database.**

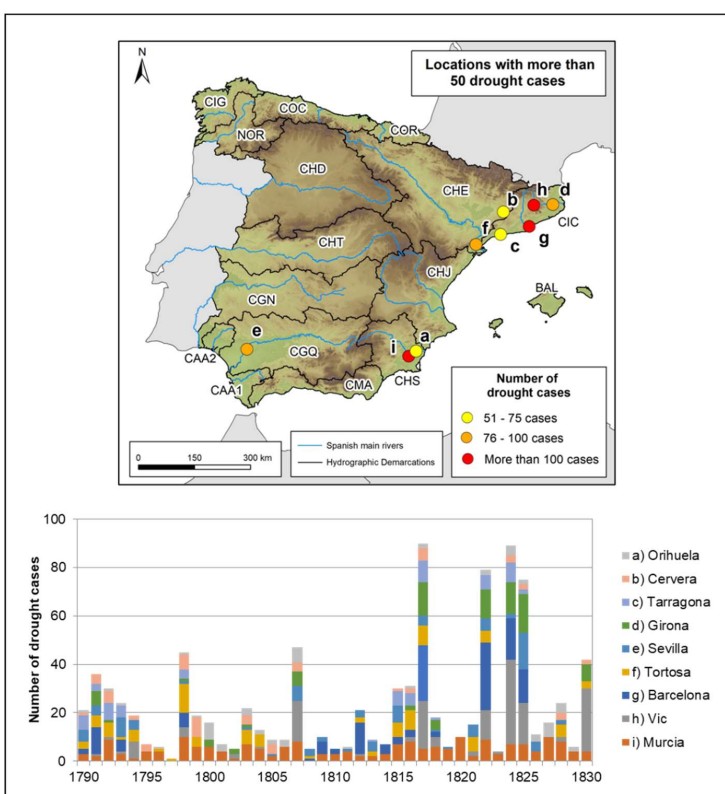

**Figure 4: Towns with more than 50 cases of drought during the Early 19th Century.**
**Elaboration from AMARNA database.**



**3.2. Drought analysis of the Early 19th Century on the Spanish Mediterranean Basin**

Table 3 shows historical data from the most severe drought episodes of the Early 19th Century based on all the cases from all the Spanish towns that record rogation ceremonies for each drought episode. In this regard, it will be possible to consider the different nuances that appear in the most representative droughts of the analysed period.

| Episode | Year of greatest impact (Nº Cases) | Approximate duration | Total cases |
|---|---|---|---|
| 1798 -1799 | 1798 (58) | 25 months | 78 cases |
| 1807 -1808 | 1807 (49) | 19 months | 55 cases |
| 1812 -1814 | 1812 (24) | 21 months | 44 cases |
| 1816 -1818 | 1817 (122) | 37 months | 175 cases |
| 1822 -1825 | 1824 (97) | 40 months | 279 cases |

**Table 3: Summary of the severe drought episodes according to historical data for the Early 19th Century. Elaboration from AMARNA database.**

The first of these episodes runs from December 1797 to December 1799, with the peak of intensity in March and April 1798. This episode stands out as it occurred several years before the megadrought of 1812-1825 and was possibly an episode still linked to Maldà Oscillation (Barriendos & Llasat, 2003). It affected five hydrographic basins (Catalan basins, Ebro, Segura, Tagus and Guadalquivir), three of which are Mediterranean (Figure 5). Despite its considerable extension, this episode had a limited duration, with only a few months of rogations. The exception is the municipality of Murcia, where rogations were recorded for 10 of the 25 months that the episode lasted. Furthermore, this episode was noteworthy in this town due to plague outbreaks (Zamora Pastor, 2001).

The second episode of severe drought occurred between January 1807 and July 1808 (Figure 5), with the largest number of cities holding rogations in October 1807. It affected six river basins (Catalan basins, Ebro, Balearic basins, Segura, Duero and Guadalquivir), four of which are Mediterranean. Its main characteristic is that it had a greater impact on towns in the southern sector of the Atlantic and Mediterranean watersheds of the Peninsula, such as Murcia and Seville.

The third episode accumulated less cases of drought but marked the beginning of the megadrought that lasted until 1825, with different regional effects throughout the sequence. It occurred between March 1812 and April 1814 with the peak of greatest severity in April 1812 (Figure 5). Despite the low number of rogations recorded (44), significant effects on crops were documented, causing wheat shortages and widespread famine in the Mediterranean basins. It had a broad impact across the Iberian Peninsula, affecting eight river basins (Catalan basins, Ebro, Balearic basins, Júcar, Segura, Duero, Tagus and Guadalquivir), three of which are in the Atlantic watershed.

The fourth episode runs between December 1815 and November 1818 (Figure 5) and stands out for the impact of the drought during 1817, which was very severe in Catalonia with instrumental records in Barcelona that were unprecedented until then (Moruno, 2021). In this episode, there was an exceptionally dry month (April 1817) in which fourteen of the twenty municipalities recorded *pro pluvia* rogations. This drought affected eight very broadly distributed river basins; four Mediterranean basins (Catalan basins, Ebro, Balearic basins and Segura) and four Atlantic basins (Galician basins, Duero,



Tagus and Guadalquivir). Rogations were made during this drought for many months, particularly in the

cities of Murcia and Girona with 12 and 11 months, respectively.

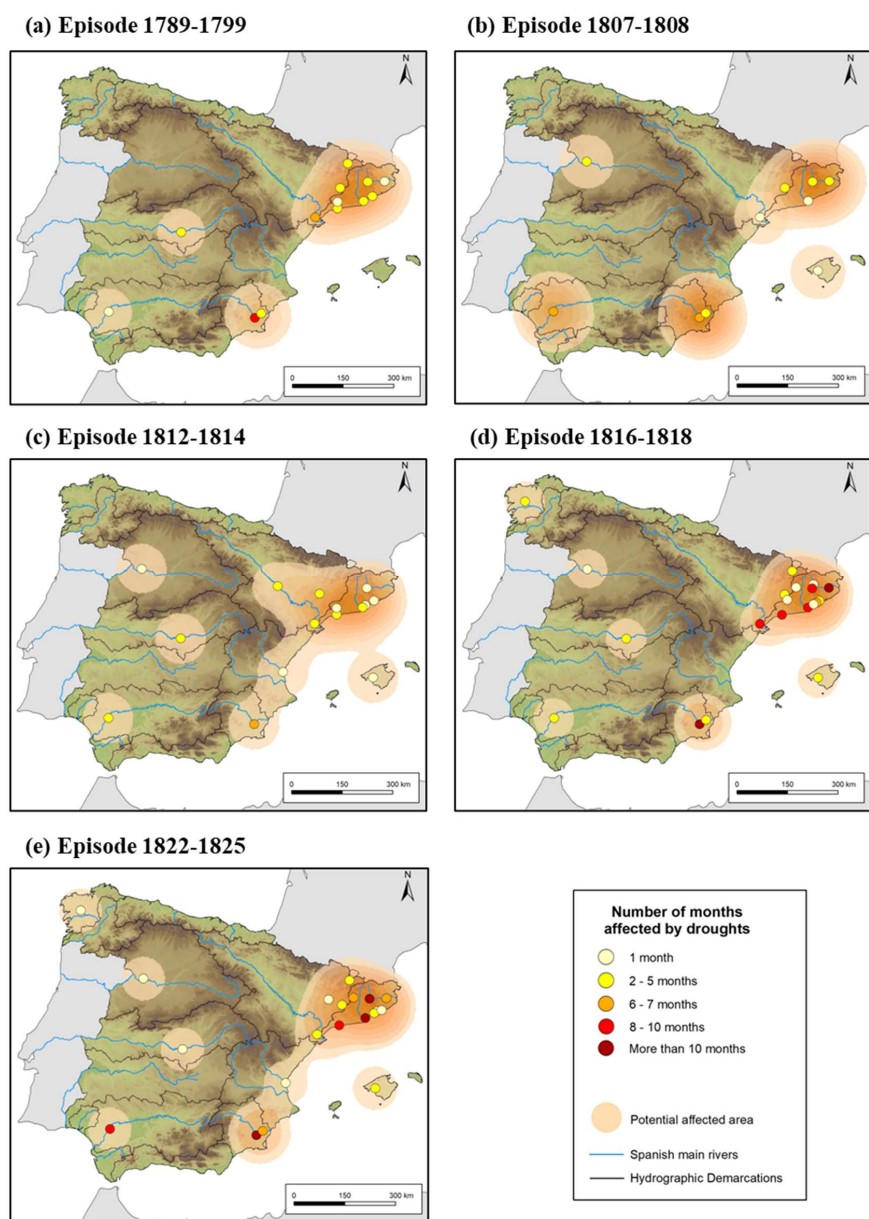

**Figure 5: Distribution of *pro pluvia* rogations by municipality. (a) Drought episode of 1798-1799. (b) Drought episode of 1807-1808. (c) Drought episode of 1812-1814. (d) Drought episode of 1816-1818. (e) Drought episode of 1822-1825. Elaboration from AMARNA database using Arc Map GIS Software, applying Kernel Density Tool.**



The last episode took place between January 1822 and January 1826 (Figure 5), although the
year 1823 recorded a low number of rogations. This drought is noteworthy for being the longest and most
persistent of the Early 19th Century (40 months). Three different peaks of severity can be observed:
March 1822, April-May 1824 and February 1825. This drought affected eight very broadly distributed
river basins: four Mediterranean basins (Catalan basins, Ebro, Balearic basins and Segura) and four
Atlantic basins (Galician basins, Duero, Tagus and Guadalquivir). Also significant was the large
accumulation of rogations carried out each month in the towns affected. For example, the town of Vic
recorded twenty months of rogations, Murcia seventeen months and Barcelona, fifteen. This episode was
accompanied by price increases of wheat and the emergence of a locust plague which affected different
towns (Azcárate, 1996).

### 3.3. Analysis of the instrumental precipitation data series of Barcelona (1786-2022)

The analysis of the instrumental precipitation data series of Barcelona (1786-2022) was
developed using three different drought indices (SPI, SPEI and Deciles) (Figure 6). The three drought
indices reported a significant number of extreme drought events, both in severity and duration, during the
Early 19th Century. A dry period between 1812 and 1825 stands out for its significant severity and
duration. The three drought records also show values of relative abundant rainfall from the end of the
19th Century until the end of the 20th Century. The beginning of the 21st Century reveals an upturn in the
severity and duration of drought episodes with respect to the 20th Century. This dry period that continues
to the present day appears to be less intense to those of the Early 19th Century, but may eventually
become of similar duration and severity.
The SPI, in comparison with the behaviour of the other two indices, highlights more clearly the
peaks of greater severity, both positive and negative (Figure 6). In this regard, 1817 stands out as the
driest year in the precipitation data series, with months of maximum severity reaching values close to -4
(-3.91 in the month of August) (Table 5). If we look at the results of this index, it becomes clear that after
the Early 19th Century, during the 1830s, the years in drought conditions were prolonged, ending around
1840. From the mid-19th Century, a new phase began with a low presence of prolonged dry periods until
the end of the 20th Century. In the 21st Century, severe drought values can be observed again. For
example, in 2021, a negative value of the SPI of close to -3 was recorded for the first time since the Early
19th Century. The SPEI shows a different result to the other two indices as it combines rainfall and
temperature values. In this respect, it is noteworthy that the most severe year of the series, according to
the SPEI was not 1817 but 1822. It is possible that the negative thermal effect of the Tambora eruption
(1815) was still significant in 1817, resulting in 1822 having a higher temperature and, consequently, a
lower SPEI value. The 1870-1890 drought episode, which does not stand out so much in the other two
indices, is also perceived as severe. With regard to the 20th Century, SPEI shows a phase of positive
values that lasted twenty years from the 1970s to the 1990s with almost not a single month with negative
values. In contrast, for the beginning of the 21st Century there are hardly any years with such positive
values (Figure 6). Undoubtedly, the recent thermal warming increases the intensity of negative SPEI
values and presents increased problems for water management.





The behaviour of the Deciles index is very similar to that obtained with the SPI index. This index
softens the extreme positive and negative behaviours. Thus, the interpretation of rainfall abnormalities do
not help, with only the most evident episodes being highlighted.

**(a) SPI – 12 months**

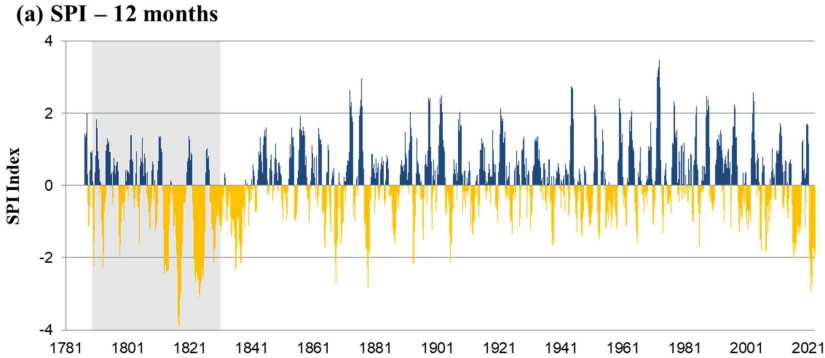

**(b) SPEI – 12 months**

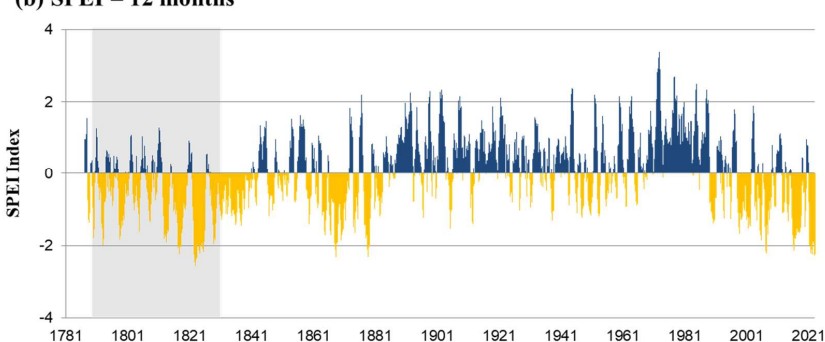

**(c) Deciles – 12 months**

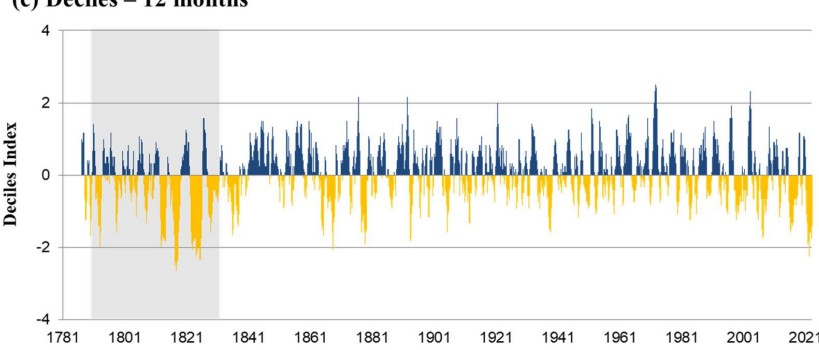

**Figure 6: Monthly values of the SPI, SPEI and Deciles indices for the Instrumental precipitation data series of**
**Barcelona (1786-2022). The study period has been shaded in grey. Elaboration with the data obtained from**
**Prohom et al., 2016.**
The results obtained with the Pettitt Test are very similar for the SPI and Decile index values,
although there are differences with respect to the SPEI index (Table 4). The main difference is the
position of the first breakpoint which, for the case of the SPI and Deciles, occurred right at the end of the
Early 19th Century, in the 1840s. On the other hand, for the SPEI index, this first breakpoint occurred at



the end of the nineteenth century, when a strong dry period ended that had lasted from 1860 to 1880 and
is much more important in this index than in the other two analysed. With respect to the breakpoint that
marks the end of the wet period of the twentieth century, the SPI and Deciles indices coincide with the
same period, at the end of 1997. Meanwhile, the SPEI marks it at the end of the 1980s, after the wet phase
of the 1970s and 1980s. From this point, the three indices go back to indicating negative averages for
their respective series (Table 4).

| Monthly data series | Pettitt Test Results | | | | |
|---|---|---|---|---|---|
| | 1rst section's average | 1rst breaking point | 2nd section's average | 2nd breaking point | 3rd section's average |
| SPI | - 0.48 (669 m: 56 yr) | October 1842 | 0.21 (1862 m: 155 yr) | December 1997 | -0.22 (301 m: 25 yr) |
| SPEI | - 0.43 (1157 m: 96 yr) | June 1883 | 0.56 (1266 m: 105 yr) | December 1988 | -0.52 (409 m: 34 yr) |
| Deciles | - 0.34 (643 m: 54 yr) | August 1840 | 0.15 (1886 m: 157 yr) | October 1997 | -0.25 (303 m: 25 yr) |

**Table 4: Results of the breakpoints according to the Pettitt Test for drought indices (SPI, SPEI and Deciles).**
**Elaboration with the data obtained from Prohom et al., 2016.**
Based on the values of the three indices, the drought episodes are summarised for the Barcelona
data series (Table 5). It reveals a greater number of drought episodes recorded in the 19th Century
compared to the 21st Century, in which the droughts were not only scarce but also less severe and shorter
(Figure 7). This can be confirmed if we consider that in the first twenty years of the 21st Century the
same number of droughts have been recorded as those occurring throughout the whole of the 20th
Century.
The droughts of the Early 19th Century period (Nr. 2 to 8) stand out due to their extreme
severity, particularly those in the central part of the period, when not only were the droughts severe but
also a large number of dry months were concentrated during this time (Table 5, Figure 7). For the rest of
the drought episodes of the series, we can observe that the majority had shorter duration (Figure 7). Only
three noteworthy drought episodes are outside of the Early 19th Century: 1877-1879 (Nr. 14), 2015-2018
(Nr. 23) and 2021-2022 (Nr. 24).

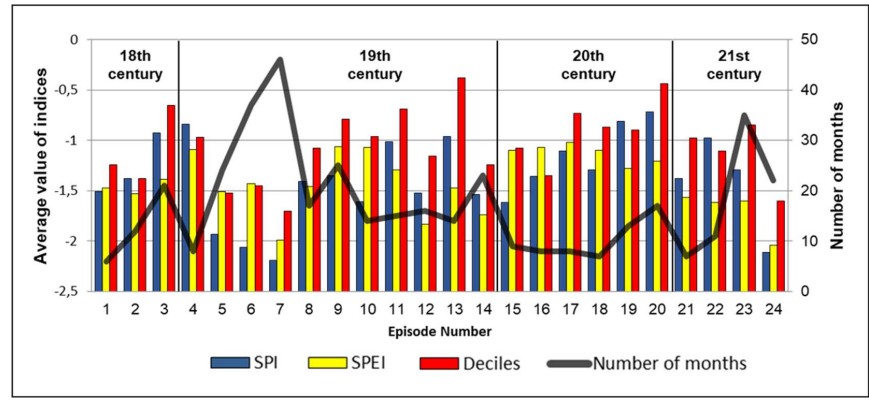

**Figure 7: Representation of the mean values of the indices and the duration in months of the drought episodes**
**described in Table 6. Elaboration with the data obtained from Prohom et al., 2016.**



| Episode Num. | Date | | Month Num. * | Averages of index values for each episode | | | Minimum values of the episodes ** | | | |
|---|---|---|---|---|---|---|---|---|---|---|
| | Onset | Ending | | SPI | SPEI | Dec. | SPI | SPEI | Dec. | Month |
| 1 | 1789/09 | 1790/03 | 6 | -1.34 | -1.47 | -1.24 | -2.22 | -1.79 | -1.67 | 11/1789 |
| 2 | 1792/05 | 1793/04 | 12 | -1.38 | -1.53 | -1.38 | -2.29 | -2.00 | -2.00 | 01/1793 |
| 3 | 1798/03 | 1798/12 | 10 | -1.36 | -1.59 | -1.02 | -1.94 | -1.76 | -1.58 | 05/1798 |
| 4 | 1807/09 | 1808/04 | 8 | -0.84 | -1.09 | -0.97 | -1.19 | -1.31 | -1.33 | 01/1808 |
| 5 | 1812/05 | 1814/05 | 25 | -1.88 | -1.47 | -1.49 | -2.46 | -1.82 | -2.00 | 10/1812 |
| 6 | 1815/11 | 1818/11 | 37 | -2.06 | -1.43 | -1.45 | -3.91 | -2.24 | -2.67 | 08/1817 |
| 7 | 1822/03 | 1825/11 | 45 | -2.23 | -2.02 | -1.75 | -3.10 | -2.22 | -2.17 | 01/1824 |
| 8 | 1828/01 | 1829/05 | 17 | -1.41 | -1.46 | -1.08 | -2.14 | -1.95 | -1.58 | 10/1828 |
| 9 | 1834/04 | 1836/04 | 25 | -1.35 | -1.06 | -0.78 | -2.35 | -1.44 | -1.67 | 11/1835 |
| 10 | 1836/11 | 1837/12 | 14 | -1.61 | -1.07 | -0.96 | -2.17 | -1.46 | -1.42 | 08/1837 |
| 11 | 1864/04 | 1864/11 | 8 | -1.34 | -1.51 | -1.11 | -1.71 | -1.72 | -1.50 | 09/1864 |
| 12 | 1867/10 | 1868/10 | 13 | -1.74 | -1.94 | -1.27 | -2.69 | -2.32 | -2.08 | 03/1868 |
| 13 | 1869/10 | 1870/07 | 10 | -1.18 | -1.60 | -0.48 | -1.62 | -1.86 | -0.75 | 11/1869 |
| 14 | 1877/05 | 1879/01 | 21 | -1.64 | -1.78 | -1.33 | -2.80 | -2.31 | -1.92 | 08/1978 |
| 15 | 1886/07 | 1887/08 | 14 | -1.13 | 0.05 | -0.57 | -1.60 | -0.16 | -0.92 | 03/1887 |
| 16 | 1893/02 | 1893/10 | 9 | -1.44 | -0.19 | -1.22 | -2.19 | -0.79 | -1.83 | 04/1893 |
| 17 | 1904/12 | 1905/10 | 11 | -1.47 | -0.99 | -0.99 | -2.15 | -1.53 | -1.58 | 04/1905 |
| 18 | 1937/11 | 1938/08 | 10 | -1.26 | -0.98 | -1.27 | -1.65 | 1.32 | -1.50 | 03/1938 |
| 19 | 1947/05 | 1948/01 | 9 | -1.07 | -1.01 | -0.67 | -1.40 | -1.23 | -0.83 | 10/1947 |
| 20 | 1952/10 | 1953/05 | 8 | -1.22 | -1.05 | -0.8 | -1.49 | -1.20 | -1.00 | 03/1953 |
| 21 | 1965/02 | 1965/09 | 8 | -1.23 | -0.76 | -0.52 | -1.58 | -0.88 | -0.75 | 09/1965 |
| 22 | 2005/03 | 2005/09 | 7 | -1.38 | -1.57 | -0.98 | -1.79 | -1.86 | -1.25 | 07/2005 |
| 23 | 2006/11 | 2007/04 | 6 | -1.28 | -1.84 | -1.46 | -1.84 | -2.15 | -1.75 | 01/2007 |
| 24 | 2015/09 | 2018/07 | 35 | -1.29 | -1.60 | -0.85 | -2.00 | -2.15 | -1.50 | 03/2016 |
| 25 | 2021/04 | 2022/12 | 21 | -2.11 | -2.04 | -1.6 | -2.92 | -2.22 | -1.92 | 09/2021 |

**Table 5: Drought episodes in the instrumental precipitation data series of Barcelona (1786-2022).**
**\*Number of months determined by the following criteria: Episodes must have at least 6 months below "-1"**
**value of SPI Index. The count of months will start and finish with the values below "-0.75".**
**\*\*The month with the lowest value of each episode corresponds to the SPI index. Elaboration with the data**
**obtained from Prohom et al., 2016.**
**4. DISCUSSION**
The comparison between the results obtained from the historical data and the instrumental data
set is part of the main objective of this study. This comparison makes it possible to contrast the reliability
of the methods used and to assess the consistency of the results obtained.
The combination of different *proxy* data expands the knowledge on the extreme
hydrometeorological events, whether they be excesses or deficits, occurring in the past. In this case, the
historical data and the instrumental data set of Barcelona have allowed us to analyse one of the driest
known periods in the study area (Table 6). The comparison of the standardised values of the historical
series with the instrumental indices enables us to observe the synchrony between the historical *proxy* and
the instrumental data (Figure 8). The coincidence of the duration of the episodes from the historical data
and instrumental sets is noteworthy. The only episode for which the durations are different is that of 1807,
attributable to the fact that it mainly affected and for longer the southern regions of the Iberian Peninsula.
In terms of the severity of the episodes, the coincidence between the two sets of data is also noteworthy,
with the episodes with most documented cases coinciding with those with a lower SPI index. The only
episode that does not follow this pattern is that of 1812, in which the number of negative ERE cases is
relatively low. But, on the other hand, according to the SPI, it is the episode with the third lowest mean of
the Early 19th Century (1790-1830). The use of elements related to the social vulnerability to drought and





extending the length of the data collection in different locations would help to resolve these specific
uncertainties and constitute lines of research to be developed in the future.

| Episod. Num. | Date according to historical data | | Month Num. | Date according to instrumental data | | Month Num. | Number of cases | | SPI episode average |
|---|---|---|---|---|---|---|---|---|---|
| | Onset | Ending | | Onset | Ending | | ERE Pos. | ERE Neg. | |
| 3 | 12/1797 | 12/1799 | 25 | 03/1798 | 11/1799 | 21 | 37 | 78 | -0.93 |
| 4 | 01/1807 | 07/1808 | 19 | 09/1807 | 04/1808 | 8 | 17 | 55 | -0.84 |
| 5 | 03/1812 | 04/1814 | 21 | 05/1812 | 04/1814 | 24 | 24 | 44 | -1.93 |
| 6 | 12/1815 | 11/1818 | 37 | 11/1815 | 11/1818 | 37 | 20 | 175 | -2.06 |
| 7 | 01/1822 | 01/1826 | 40 | 01/1822 | 10/1825 | 46 | 41 | 279 | -2.19 |


**Table 6: Characteristics of the five principal drought episodes of the Early 19th Century (1790-1830) according to historical data and instrumental series. Elaboration with the data obtained from Prohom et al., 2016 and the data from the AMARNA database.**

Figure 8 shows the coincidence of the droughts according to the historical data (positive values)
with the negative oscillations shown by the SPI and SPEI indices. The overlapping of this information
highlights the importance of the droughts in the final part of the Early 19th Century, specifically between
1815 and 1825, although the instrumental data indicate that this period could have started in 1812. For
this reason, it is desirable to analyse in more detail the three drought episodes in which there is a high
degree of alignment between the instrumental data the historical proxy data:
- That of 1798 stands out as it forms part of the rainfall irregularity typical of the Maldà
Oscillation (Barriendos & Llasat, 2003). This drought occurred between two phases of
intense rainfall. The alternation of floods or heavy rains with droughts is typical in areas
with Mediterranean climate. Despite this fact, this is the only drought of the Early 19th
Century which precedes and is preceded by flood or heavy rains episodes in the Northeast
Iberian Peninsula.
- The drought of 1817 was different as it was the most severe according to the SPI index and
was the year during which the most drought cases were recorded in the whole of the Early
19th Century (120). Despite this strong impact, mainly corresponding to the first half of
1817, the episode was not as long as that of 1822-1825 and, for this reason, according to the
SPEI index, it was less severe than this latter episode.
- That of 1822-1825 stands out for its duration of around 40 months according to the
rogations and 46 months according to the instrumental series. This not only makes it the
longest episode of the Early 19th Century but also of the whole of the precipitation series of
Barcelona (Table 6). Moreover, this episode is the one with the highest severity, both in
terms of the accumulation of drought cases (279) and in terms of the SPI average for the
episode as a whole. It is also worth mentioning that according to the SPEI index, this
episode is the most severe of the entire Barcelona rainfall data series.
Based on the standardised data series of the number of droughts for the Early 19th Century, the
correlation coefficient has been calculated with the values of the different drought indices, with which the
precipitation sets of Barcelona have been analysed (Table 7). We can observe that the values of the three
indices generate correlations of over -0.5; where the Deciles index has the greatest correlation value,
which is inverted (-0.65). The same is confirmed for the coefficients of determination: it verifies that




between 35% and 42% of the variability is explained by the behaviour of the variables used (drought
indices vs. number of drought cases).

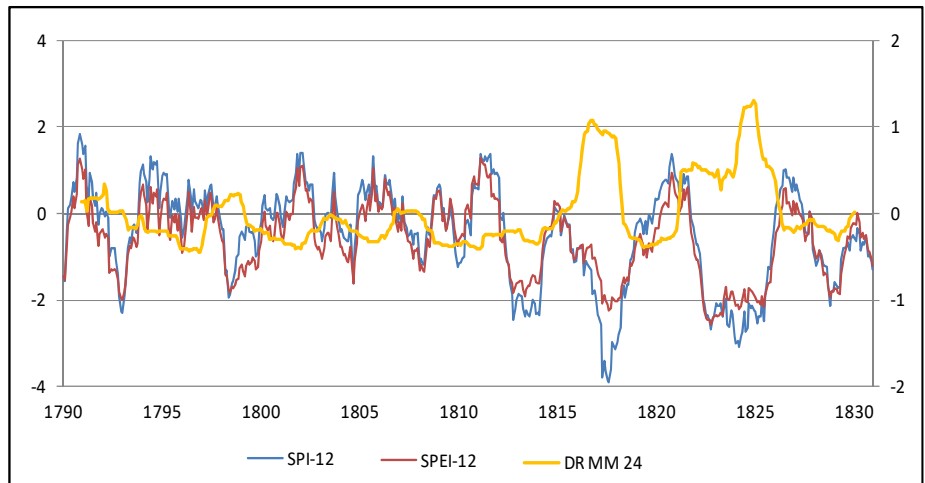

**Figure 8: Comparison of the results of the drought indices (SPI and SPEI) and the two-year moving average of**
**the standardised monthly values of the drought cases (DR). Elaboration with the data obtained from Prohom**
**et al., 2016 and the data from the AMARNA database.**

| Index | Correlation coefficient (R) | Correlation coefficient ($R^2$) |
|---|---|---|
| SPI | -0.62 | 0.38 |
| SPEI | -0.59 | 0.35 |
| Deciles | -0.65 | 0.42 |

**Table 7: Correlation and determination coefficients.**
The study of extreme drought episodes in the past is important for understanding the pattern of
low frequency episodes and for addressing the droughts occurring in the context of climate change, which
have erratic behaviour according to the most recent models (IPCC, 2021). Furthermore, the knowledge
generated for the study over a long period of time also enables us to better understand the vulnerability of
society in different historical contexts and the way in which it has adapted over time to droughts.
Different studies carried out on droughts for the whole of the Mediterranean region for long time
periods (Kim & Raible, 2021; Xoplaki, et al., 2018; Marcos-García, et al., 2017) reveal that it is one of
the most vulnerable regions to this natural risk within the context of global warming. Taking into account
the results of this paper, the importance of droughts in the Mediterranean region are underlined, it is
necessary to have the support of different drought indices, as well as other climatic indicators to
determine their severity (Kim & Raible, 2021). The availability of older instrumental data series is highly
important to find a wider range of drought severities and typologies than those found only by analysing
the 20th Century. This relationship is evidenced in the research carried out by Erfurt et al., 2020 that
combines historical instrumental data with dendrochronological records to analyse the period of the
beginning of the 19th Century in south-east Germany. With respect to the use of dendrochronological
data to analyse the droughts and megadroughts of the past, the Old-World Drought Atlas is also worth
mentioning (Cook et al., 2015). This publication includes a severe drought that occurred at the beginning
of the 19th Century, between the Little Ice Age and the Modern Climate period.



Other authors, particularly in the study of the Iberian Peninsula, have used historical data for classifying droughts in the period at the beginning of the nineteenth century (Domínguez-Castro et al., 2012; Gil-Guirado et al., 2019; Gil-Guirado & Pérez-Morales, 2019). It is worth highlighting the article by Domínguez-Castro et al., 2012, in which the historical data is combined with instrumental data to characterise the droughts of the period analysed in Spain. In this case, the same dry periods of great intensity are detected (1817 and 1824) by both the historical and the instrumental data series The authors conclude that the relationship between these droughts and external forcing factors is clear, but more research is also required to confirm it.

Furthermore, the modelling used by (Kim & Raible, 2021) does not show any extraordinary occurrence of droughts for the Mediterranean region as a whole during the Early 19th Century. Neither do these authors relate rainfall patterns with that of those volcanic eruptions emitting more particles into the lower stratosphere, such as Tambora. According to their study, droughts occurring in the Mediterranean are due mainly to the internal dynamics of the climate system and not to external forcing factors (inter-tropical volcanic eruptions and solar radiation variations) (Kim & Raible, 2021). The same conclusion has been obtained for the Eastern Mediterranean region, although for another period than the one studied in this research (Xoplaki, et al., 2018). For these reasons, it may be concluded that the relationship between the external forcing factors can lead to different rainfall pattern depending on the region in which specific conditions prevail.

In the case of the Iberian Peninsula, the combination of inter-tropical volcanic eruptions with positive phases of the North Atlantic Oscillation during the first two years after the eruption could result in dry periods for the Iberian Peninsula and in wet phases for Central Europe (Domínguez-Castro et al., 2012). To that, the lack of droughts detected in south-east Germany during the Early 19th Century could reinforce this hypothesis (Erfurt et al., 2020). In this study of droughts for south-east Germany, despite the lack of droughts in the Early 19th Century, there were temporal coincidences with other severe drought episodes, such as those occurring at the end of the 19th Century (between 1857 and 1870) and at the beginning of the 21st Century (2003 to 2018) (Erfurt et al., 2020). This period coincides with two of the most severe episodes of this century according to the records of the instrumental precipitation data series of Barcelona: the drought of 2007-2008 and that of 2015-2018 (see Table 5).

**5. CONCLUSIONS**

The results obtained with this broad time-scale research contribute to a better understanding of drought episodes occurring in the early 21st Century in the study area. Data collection and extension of databases, allows a substantial improvement of knowledge about drought patterns in the study area.

One of the main results achieved in this research is the high negative correlation between the drought historical data and the instrumental precipitation data sets of Barcelona. This correlation validates the historical information for the study of climate droughts in historical perspective. Despite their different origins and methodologies, these two data sources have shown that they can provide information that is comparable, enabling the reinforcement of the importance of the episode recorded, either floods or droughts.



The combined use of instrumental and historical sources shows changes in rainfall variability in
specific periods, alternating between periods of heavy rainfall and drought. Accordingly, in the Spanish
Mediterranean basins during the Early 19th Century, between1810 and 1830, the alternation between
periods of heavy rainfall and drought is revealed. In contrast, rainfall patterns during the preceding
climatic phase of the Maldá Oscillation (1760-1800) were directly opposite to those observed during the
Early 19th Century. Additionally, the analysis of instrumental data shows the similar pattern of severe
droughts between the end of the LIA and the current context of Global Warming. On the other hand, the
20th Century does not show such a pattern of severe droughts.
The integration of historical documentary sources with instrumental records for identifying
severe droughts has yielded promising outcomes. This methodology, leveraging documentary evidence,
has been proven viable for periods or regions lacking instrumental data. Building on the success of
merging these two climatic information sources, a prospective research direction for the Early 19th
Century and other significant climatic epochs involves amalgamating historical data with evidence from
other climatic proxies, particularly dendrochronology, alongside instrumental pressure series. Such an
approach would enhance our comprehension of the atmospheric processes at a synoptic scale, elucidating
the mechanisms behind the most severe drought episodes.

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

**AUTHOR CONTRIBUTION**

Josep Barriendos: Data processing and analysis. Interpretation of the results. Preparation of graphic and cartographic material.

María Hernández Hernández: General revision of the texts and advice on the preparation of the materials.



Salvador Gil-Guirado: Methodological approach and advice on the conceptual criteria for defining
drought.
Jorge Olcina Cantos: General review and advice on the conceptual criteria for defining drought.
Mariano Barriendos: Elaboration and organisation of information from historical sources.

**COMPETING INTERESTS**
The contact author has declared that none of the authors has any competing interests.

**ACKNOWLEDGEMENTS**
This article has been possible thanks to the support and finance from:
- "Ayudas para estudios de Máster oficial e iniciación a la investigación (AII)", University of Alicante.
- "Ajuts Joan Oró per a la contractació de personal investigador predoctoral en formació (FI)", year 2023.
Government of Catalonia.
- "Grupo de investigación de Agua y Territorio", University of Alicante.
- "Grupo de Clima y Ordenación del Territorio", University of Alicante.
- "Departamento de Análisis Geográfico Regional y Geografía Física", University of Alicante.