# Peer review of "Droughts of the Early 19th Century (1790-1830) in"

_EGUsphere, 2024_

## Referee Comment (RC2)

[referee-annotated manuscript omitted]

---

## Author Response (AR1)

Response to the reviewers' comments

We have attached below the responses to the comments and suggestions made by the reviewers. We have added the line reference on each of the responses to the reviewers' comments.

From this point on, the comments of the reviewers are written in black and the answers of the authors to these comments are written in red.

**Author's response to RC1: 'Comment on egusphere-2024-832', Anonymous Referee #1, 16 May 2024**

Answer:
The authors express their gratitude for the suggestions made in this review by Referee #1. In the following text, we provide a detailed answer to all those comments that have been suggested by Referee #1.

Comment:
This study presents reconstructions of droughts of the early 19th century in the Northeast Iberian Peninsula. The focus lies on historical data (Rogation ceremonies), but instrumental data is also used. The manuscript is well-written and well-structured, which leads to a good understanding of the presented content.

It begins with an extensive introduction, in which terminologies are explained and the climate of the Iberian Peninsula is explained to argue for the relevance of investigating droughts from 1790 to 1830. Furthermore, the introduction includes a research background that shows the climatological characteristics of this period. Moreover, the relevant literature is elaborated carefully and followed by the research objectives.

The section "Materials and Methods" introduces the historical data taken from the AMARNA database and the instrumental data (precipitation measurements from Barcelona). In this context, the indexation of the historical data and the calculated climate indices of the instrumental data (SPI, SPEI, and Deciles) are well explained.

The result section is well presented and shows the results of instrumental and historical data in separate subsections. In terms of the historical data, the temporal distribution of positive EREs and negative EREs are analyzed, as well as the geographical distribution over the Iberian Peninsula. The authors further detected five episodes of severe droughts in the period of interest. This is followed by presenting the three indices SPI, SPEI, and deciles, which were calculated from the instrumental precipitation data from Barcelona. Thereby, the interest also lies in the time series and detecting drought episodes.

The discussion section compares the two different sources of reconstructing droughts and shows a high agreement. This shows that rogation ceremonies are a good proxy for droughts. The conclusion section summarizes the results accordingly.

This paper adds value to the scientific literature by compiling and describing the monthly variability of extreme hydrometeorological events in the Iberian Peninsula, identifying drought episodes, and combining/comparing historical data and instrumental data. The results are not restricted to the Northern Iberian Peninsula and the period of 1790 to 1830 and also include historical data of the whole Iberian Peninsula as well as the whole time series of the precipitation measurements of Barcelona (1786–2022).

**I suggest accepting this study with only a few minor revisions.**

I also added optional minor revisions, which might be changed if the second reviewer also mentions them.

**Minor revisions:**

Comment:
Fig. 1 and Fig.3: The legend is hardly readable in my A4 printout.
Answer:
Thank you for this suggestion. We have modified the legend of these and others figures in order to make the legends more readable (See Lines 435 (Figure 1) and 534 (Figure 3) of the revised manuscript).

Comment:
Line 19: I would write something like "crucial" instead of "imperative."
Answer:
Thank you for this suggestion. We have made the change suggested by the reviewer (See Line 19 of the revised manuscript).

Comment:
Line 26: "of this century" has the potential to be misunderstood (although understanding it as the 21st century wouldn't make sense). Anyway, write maybe "…were more frequent and severe during the Early 19th century compared to the Late 19th century."
Answer:
Thank you for this suggestion. We understand that this expression can be misunderstood and we have changed it by the suggested sentence (See Lines 26-27 of the revised manuscript).

Comment:
Line 188: Without reading the given source, it doesn't get clear if there were, in total, 247 eruptions or if 247 eruptions had a VEI of [2, 3).
Answer:
Thank you for this suggestion. We have rewritten the text so it can be more clearly explained (See Lines 209-210 of the revised manuscript).

Comment:
Line 239: First use of the acronym "AMARNA". Thus, it might be introduced/explained here and not in line 252.
Answer:
We appreciate and accept the comment, as the AMARNA database is a fundamental part of the work it should be mentioned earlier in the text. Currently the first mention of AMARNA is in the Introduction (See Lines 179-187 of the revised manuscript).

Comment:
Line 420: Based on the data in Fig. 2, I would change "1800" to "1805".
Answer:
Thank you for this suggestion. The change has been accepted and added into the text (See Line 501 of the revised manuscript).

Comment:
Lines 566 to 571: I am not sure how "noteworthy drought episodes" are defined here. Is the definition dependent on the SPI, the SPEI, the Deciles, or on all three? According to Fig. 7, it is hard to see how Nr. 23 is more noteworthy than, for instance, Nr. 21 or Nr. 22.
Answer:
We understand that the current sentence can led to misunderstanding, so we have rewritten and modified the sentence for these lines (See Lines 657-659 of the revised manuscript):

"According to the average value of indices and the number of dry months only three noteworthy drought episodes are outside of the Early 19th Century: 1877-1879 (Nr. 14), 2015-2018 (Nr. 23) and 2021-2022 (Nr. 24)."

**Optional minor revisions:**

Comment:
Line 348: It might be interesting to introduce the different types of ceremonies.
Answer:
Thank you for this suggestion. We will change those lines in order to explain rogation ceremonies in a clearer way. We will also differentiate between the *pro pluvia* ceremonies and the *Te Deum Laudamus* ceremonies which represent the end of the drought event (See Lines 394-408 of the revised manuscript).

Comment:
Introduction: Although the Authors substantially introduce the existing literature and the study's objectives, it isn't clear to me how this study differs from earlier studies analyzing droughts in the Iberian Peninsula. Suppose the answer is that it is the first study with an increased resolution (monthly) of the historical data and with qualitatively and quantitatively improved historical data. In that case, this point can be ignored. Also, this period might not be investigated in such detail.
Answer:
Exactly, this study represents the first effort in the Iberian Peninsula to analyse drought episodes at a daily resolution whereas previous efforts used information at yearly o seasonal resolutions.

**Author's response to RC2: 'Comment on egusphere-2024-832', Marzena Kłusek, 11 Jun 2024**

Answer:
The authors express their gratitude for the suggestions made in this review by Referee #2. In the following text, we provide a detailed answer to all those comments that have been suggested by Referee #2.

Comment:
The reviewed manuscript presents an analysis of drought periods (their frequency and intensity) in the Iberian Peninsula between 1790 and 1830 CE. The research was carried out by comparing meteorological data and historical records. Used historical documentary sources show the daily to monthly occurrence of sacral ceremonies (rogation) which allows research to be carried out with a high temporal resolution.

Comment:
Manuscript is well written in terms of content and text subdivision. It contains a thorough description of the climatic conditions prevailing in the study area and references to similar studies conducted here in the past. Article has a well-defined aim and scope of the work. The results obtained are clearly presented and visualised in graphs and tables as well as compared with the findings of previous studies. However, this comparison and discussion could also be extended to include the effects of analyses carried out using other proxies (for example, dendrochronological and lake sediments), which would significantly increase the value of the text. In addition to this, the methodology of the study needs to be completed, and should include a more detailed explanation of how the individual coefficients (SPI, SPEI, Deciles) were calculated and which computer programs were used to conduct the analyses.

Answer:
We understand and appreciate this commentary on the underlying approach to the use of different proxy dates in a study of recent palaeoclimatology. The team of co-authors has considered to present at this level of scientific article the results of the documentary proxy only with the support of early instrumental period data for two reasons: firstly, the large amount of available material which made it advisable not to extend the work with other associated lines of analysis. Secondly, and perhaps more importantly, the availability of proxies for pluviometric reconstruction is not yet widely available in the region studied. For instance, we know of dendroclimatic and limnological studies, but they analyse a high mountain area (Pyrenees Mountains) that is not suitable for our study. They correspond to unique natural ecosystems, and probably have different environmental and climatic dynamics from the lowlands and agricultural plains where we obtain the documentary proxies.
Regarding the explanation requested on the generation of the precipitation indexes, the co-authors have considered that these are methodological procedures that have been applied in many different scientific publications for several years. For this reason, we consider that providing the bibliographical references used is quite explanatory.

Comment:
It is also worth considering the time interval chosen for the analysis. The years 1790-1830 CE are a period for which the weather record is of somewhat lower quality, due, for example, to a different methodology of meteorological measurements than today. In addition, it would be very interesting to increase the time range of the analyses carried out, as much longer instrumental record as well as historical documentary sources are

available. Especially since the analyses of meteorological data presented in the article were performed up to the year 2022 CE. Such an extension of the time frame would also give more statistically reliable results.

Answer:

We appreciate this interesting consideration. Without a doubt, in future work, we will explore the possibilities of using the proposed drought indices to correlate recent drought situations with the generated social impact. We also take into account that observations from old instrumental records may not be of the same quality as recent observations. Nonetheless, numerous research studies have validated the long series from Barcelona for the study of climate variability since the late 18th century (Domínguez-Castro et al., 2014; Prohom et al., 2016; Rodríguez et al., 2001).

On the other hand, the main objective of this work "is to examine the occurrence and magnitude of extreme droughts lasting over a year in the Spanish Mediterranean Basin during the Early 19th Century (1790-1830)." We have historical social impact data (mainly rogations) that can be correlated with early instrumental data. This is important and novel for three main reasons, which motivate focusing this work on such analysis: 1) the existence of long instrumental records, like that of Barcelona, is scarce and spatially dispersed in Spain; 2) social changes during the 19th century make historical records of social impact, based on rogations, demonstrably inconsistent after 1836 (Espín-Sánchez & Gil-Guirado, 2022), which discourages their use as proxies after this date; 3) the available instrumental records and social impact data are contemporaneous. These considerations, especially the second one, advise against extending the analysis period. However, to clarify this point, we have added part of the previous response to the revised version of the manuscript (See Lines 166-171 of the revised manuscript).

However, it should be emphasised that the reviewed manuscript is valuable and worthy of publication. The article presents very interesting research results and is of interest to a wide audience. It proves that accurate palaeoclimatic reconstruction based on historical proxies is possible. The findings of the paper, due to the high correspondence between climatic and historical data, allow the application of the developed research methodology to reconstruct the occurrence of drought periods (their frequency and intensity) in the past reaching back many hundreds of years. This is of great value. The results obtained during the study are particularly noteworthy in the context of ongoing climate change, as they allow comparison of the occurrence and magnitude of droughts between 1790 and 1830 CE with droughts appearing in later periods and especially in the last decades.

Other detailed comments are contained in the attached file.

Answer:

The authors express their gratitude for the suggestions made in this review by Referee #2. We respond below to the comments added in the attached text

Comment:
Lines 99-100: But "summer is always the season with the lowest rainfall contribution" - so not rainy season.

Answer:
This sentence has been rewritten by adding that the rainy seasons are autumn and spring (See Lines 91-100 of the revised manuscript).

Comment:
Line 188: volcanic explosivity index (VEI)

Answer:
We have added the meaning of VEI into this line as indicated by the referee (See Lines 209-210 of the revised manuscript).

Comment:
Lines 201-203: This sentence needs to be improved in terms of specifying the area.
Answer:
Thank you for your comment. The sentence as written may lead to confusion and has been modified (See Lines 224-226 of the revised manuscript).

Comment:
Lines 208-212: This fits better into the Results chapter.
Answer:
We appreciate your comment. We have followed your suggestion by removing this phrase from the introduction and placing it into the latter sections of our research (See Lines 489-493 of the revised manuscript).

Comment:
Lines 259-260: This fits better into the Results chapter.
Answer:
We consider that this phrase expresses a type of climatic characteristic typical of the Mediterranean, but which still requires a great deal of research. As the present work aims to deepen and improve the knowledge of this irregular rainfall characteristic of the Mediterranean climate, we consider appropriate to mention this circumstance in the objectives of the study itself.

Comment:
Lines 271-272: An explanation of what these indexes mean would be very useful.
Answer:
We appreciate the comment. We have added the full name of the indices (See Lines 295-296 of the revised manuscript). The explanation of the different indices is given later in the text (See Lines 441-460 of the revised manuscript).

Comment:
Lines 329-331: Please explain why data from the Fabra Observatory was not used during this period. This would increase the homogeneity of the data.
Answer:
For this study we use the official and homogenised series from published by Prohom et al., 2016. Using data from the Fabra Observatory for the years between 2014 and 2022 would not be homogenised with respect to the previous years in the series. Due to the distance between the Fabra Observatory and Barcelona, we prefer to use data from the city centre, as explained in the text.
We try to return to the locations typical of the study period (1790-1830) instead of continuing with the series from the Fabra Observatory for the following reasons: it corresponds to an altitude (400 meters above sea level) different from that of the area of plain and, secondly, began to record measurements in 1913. Between the distance from the centre of the city of Barcelona and the considerable altitude of the Fabra Observatory, we consider that the high irregularity of the precipitation in the Mediterranean climate does not recommend the use of a landmark as different as the area of the historic centre of the city of Barcelona. These considerations have been the subject of debate for years when defining climate instrument series for Barcelona. As proof of this debate, which is

still open, we can comment that the Spanish Meteorological Service has recently decided that the reference meteorological series for Barcelona should be the Observatory of the International Airport of Barcelona El Prat, located 18 km from the historic centre of Barcelona, but at a similar altitude above sea level as the urban centre.

We have added more explanations on the Fabra Observatory (See Lines 356-363 of the revised manuscript)

Comment:
Line 359: Please define it more precisely.

Answer:
We agree that the amount of data available in the AMARNA database at the time of this study needs to be clarified. We have added this information on the lines indicated by the reviewer (See Lines 415-416 of the revised manuscript).

Comment:
Lines 371-372: Please explain the abbreviations also here.

Answer:
We appreciate the comment. Adding the complete names of the river basins will improve this text (See Lines 428-429 of the revised manuscript).

Comment:
Lines 384-397: When describing these coefficients, it would be helpful to add the formulas by which they are calculated and the ranges of indexes.

Answer:
The paper now summarises the functioning of the indices and provides recently updated references for the reader to further explore them. We also added the ranges of the three indices, as this is useful information for the reader (See Lines 441-460 of the revised manuscript).

Comment:
Lines 398-400: Please extend the description of the statistical methods significantly and please specify which computer programmes were used during the calculations.

Answer:
We agree with this comment. The explanation of the techniques of statistical analysis of the data have been expanded to add the information proposed by the referee. The calculations were carried out using a set of R scripts (See Lines 461-470 of the revised manuscript).

Comment:
Lines 401-405: This description is insufficient and very unclear, please elaborate it significantly and specify all the ranges of SPI index.

Answer:
We appreciate the comment. We rewrite the paragraph from lines 401 to 405, where we explain the criteria to define the drought episodes found from the instrumental precipitation series of Barcelona. Revised text in lines 472-480 of the revised manuscript:

*Based on the results obtained from various drought indices, a detailed criterion has been established to classify the different drought episodes identified for the early 19th Century. This criterion relies mainly on the SPI to define each episode, based on the drought thresholds defined by the literature (McKee et al., 1993). Specifically: The start and end of a drought episode are determined by SPI values that cross the threshold of -0.70. This threshold is chosen to capture the transition*

*between drought episodes. Also noticing the transition periods into and out of drought conditions. A drought episode is characterised by having at least five consecutive months in which SPI values are consistently below -1.0, indicating moderate to severe drought conditions. By defining these specific ranges, we ensure a systematic and reproducible approach to identifying and analysing drought episodes.*

Comment:
Lines 433-434: Is this sentence correct and necessary?
Answer:
We appreciate the suggestion. The sentence indicated in lines 433 and 434 is somewhat redundant and can be deleted without compromising the overall sense of the figure 2 caption.

Comment:
Line 488: How does this relate to the Tambora volcano eruption?
Answer:
The text in line 488 makes no mention of the eruption of the Tambora volcano, only of the effects of the 1817 drought. In general, throughout the text we avoid to relate the 1817 drought to the 1815 eruption. The relationship between the 1815 eruption and the atmospheric conditions that may have caused this drought are still being under research.

Comment:
Line 542 (Figure 6): Why was a 12-months period chosen?
Answer:
In lines 443-446 of the revised manuscript, in the methodology section, a sentence has been added explaining the reasons why it has been chosen to analyse the instrumental series based on 12-month SPI. Due to the irregular distribution of precipitation throughout the year in the north-east of the Peninsula, the 12-month groupings are the ones that best group and detect drought episodes. Other groupings such as 3 or 6 months can detect a lack of precipitation that is typical of the usual conditions of the seasons and intra-annual variability (Gil-Guirado & Pérez-Morales, 2019).

Comment:
Line 558 (Table 4): Please explain in more detail the content of each column in the table.
Answer:
We appreciate the comment. A more detailed description of the content of each column has been added in the table caption (See Lines 642-646 of the revised manuscript).

Comment:
Line 563: Does this confirm the previous sentence?
Answer:
Thank you for your comment. We have detected an error in line 562, where instead of talking about the 21st century it should refer to the 20th century. The sentence in line 563 confirms that the 20th century has had fewer droughts and these were less severe than those of the 19th century and those of the 21st century (See Lines 650-652 of the revised manuscript).

Comment:
Line 571: 24
Answer:

We are very grateful for these comments because they have allowed us to see that figure 7 was incomplete as it was missing one of the drought episodes. It has now been updated so that both Figure 7 and Table 5 contain the same information.

Comment:
Line 571: 25
Answer:
This problem has been attended and answered with the previous answer.

Comment:
Line 572 (Figure 7): 25
Answer:
This problem has been attended and answered with the previous answer.

Comment:
Line 574: 5
Answer:
Thank you for your comment. We have changed the figure number to address the mistake.

Comment:
Line 633 (Figure 8): Axis description missing.
Answer:
We appreciate the comment and we have added labels on both axes in Figure 8.

Comment:
Line 637 (Table 7): Please elaborate on the Table caption.
Answer:
We agreed with the reviewer's suggestion. We have added further description on the table caption.

Comment:
Line 646-648: This sentence needs improvement.
Answer:
We agreed with the reviewer's suggestion. We have rewritten this sentence in order to make it clearer (See Lines 757-760 of the revised manuscript).

We greatly appreciate your meticulous review and insightful comments. Your detailed feedback and expert knowledge on the subject have significantly strengthened our manuscript. Thank you for your invaluable contribution to enhancing the quality of our work.

**Author's response to CC1: 'Comment on egusphere-2024-832', Cary Mock, 20 Jun 2024**

Answer:
The authors express their gratitude for the suggestions made in this review by CC1. In the following text, we provide a detailed answer to all those comments that have been suggested by CC1.

Comment:
Overall, I think this is a good paper eventually publishable. It presents some interesting historical climate information for the early 1800s portion of the Little Ice Age. However, I see some things that need to be revised not just in context from a historical climatologist, but also to make things more appealing and clear to a general audience for Climate of the Past. The paper reads easier in the second half of the manuscript in the results. Some points are below.

Comment:
1 - The first 154 lines is a bit of a broad literature review on drought, climate classification, historical data, etc. that can be much condensed or really eliminated and small parts incorporated later in the paper. Section 1.2 can be condensed too, as it reads more like a thesis/dissertation. It is best for the authors to get to the main points (which really started on Line 155) on what they are doing in the paper, and not ramble too much on broad aspects. One does not get an idea what the paper is really about until Line 247. Lines 273-275 that describe the main paper intentions come quite late.
Answer:
We welcome this suggestion. Taking into account the reviewer's suggestion we have synthetized the first sections of the introduction. Also, we have mentioned the main topic of the article before in the introduction section so that it is not discussed so late.
For section 1.2, it could certainly be streamlined to make this section more readable and clearer. However, other reviewers have pointed out the importance of this section and its content in order to have a clear idea of the drought problem in the study area, demanding additional bibliography on the subject. Thus, in order to make both requirements compatible, we have improved the narrative flow of this section, cutting out repeated words or concepts, while adding new bibliography on the subject.

Comment:
2 – The number of cases of data in AMARNA, the resolution whether it is monthly, seasonal, annual, daily, are explained a few times later in the paper but should be mentioned up front when AMARNA is mentioned.
Answer:
We appreciate this comment. Other reviewers also suggested that information on the AMARNA database should appear earlier in the text.
We have prepared a paragraph to be inserted in the introduction chapter (Lines 179-187 of the revised manuscript).

Comment:
3 - I am pleased the authors consider categories and subcategories of the AMARNA rogation data. However, I don't clearly understand comprehensively how a thorough

analysis on how the subjectivity in historical rogation data climate was dealt with and to relate to SPI, etc. The authors mostly cite Barriendo papers and assume that justifies in itself. A few specific examples of the data on how this is done would greatly clarify things. A verification procedure based on Table 1 criteria could also be done.

Answer:

For the case of methodological explanations concerning the treatment of historical data, Barriendos' reference covers this question (Martín-Vide & Barriendos, 1995). Also, in order to properly explains these matters in this text, it would require too much space and it is not the aim of this paper. The relationship between historical data and precipitation indices is also explained in the text. For the present study, these data are merely compared in parallel.

The procedure of indexing rogation ceremonies in levels is a climate reconstruction technique that has been validated to reconstruct water deficits. Not only in Spain, but also in many European and Latin American countries (Domínguez-Castro et al., 2021). In this respect, the work of Martín-Vide & Barriendos (1995) is the seminal work that lays the methodological foundations. In this sense, the objectives of this work do not include testing the validity of this consolidated methodology. However, we specify additional references where the methodological process for generating proxy data through rogations ceremonies can be studied in greater depth (in this regard, see the works of (Alcoforado et al., 2000; Espín-Sánchez & Gil-Guirado, 2022; Gil-Guirado et al., 2019). (Lines 396-397 of the revised manuscript).

Comment:

4 = ERE and Drought results (Figures 3 and 4) look good. I like the SPI Barcelona but perhaps more can be described regarding homogeneity, can any homogeneity test be done? SPI and SPEI results are interesting in comparison for the early 19th century.

Answer:

We appreciate the suggestion. We have expanded the description of the Methodology regarding the precipitation series, in order to further explain the series homogeneity (Lines 356-368 of the revised manuscript).

In relation to the question of whether any homogeneity test has been applied, the answer is No. Data already validated by the *Servei Meteorològic de Catalunya* (Meteorological Service of Catalonia) has been used. These data have already undergone a homogenisation process carried out by the staff of this institution.

We have chosen to use recent periods of series that have also been valid. As precipitation is a phenomenon with a high degree of spatial irregularity, we have preferred not to repeat the application of the homogenisation procedure. The data used in any case has been validated by a meteorological service that offers them in open access on-line. We therefore consider them to be fully usable.

Comment:

5 - Table 7. Have the authors considered non-parametric correlations?

Answer:

Thank you for your comment. We have performed tests to analyse the normality of the data series. Based on the results obtained, we have chosen to add the non-parametric correlation text results to the Pearson correlation results. In this sense, we have added the following specifications in the discussion section (Lines 723-730 of the revised manuscript):

*To correlate our drought index with the SPI, SPEI and Deciles values, we performed different correlation tests with RStudio software (Posit team, 2024), taking into account the normality or non-normality of the data. In this regard, the results of the Shapiro-Wilk test show that the SPI and Deciles series do not deviate significantly from normality (p-value> 0.05). However, the SPEI series and our drought index, show significant deviations from normality (p < 0.05). Given these results, we opted to apply different correlation methods: Pearson's correlation for normally distributed data, Spearman's and Kendall's correlations for data that did not meet the assumption of normality.*

The new results have been added to Table 7 (See below). In addition, we have corrected an error in the original table and replaced Correlation coefficient ($R^2$) by Coefficient of determination ($R^2$). The added results show significant differences between the different correlation tests. With a notable decrease in the R and $R^2$ values, for the non-parametric tests, with respect to Pearson's correlation. Nevertheless, the values of all the tests confirm the sign of the correlations and give robustness to the comparison of our drought index with meteorological drought indexes, especially in the case of the comparison with the deciles index.

In addition, in later phases of the investigation, when there is a better availability of historical and instrumental data, we will be able to focus on comparative analyses, correlations and even calibrations of proxy series. In any case, this will only be applicable and results can be obtained for the second half of the 19th century and the first decades of the 20th Century. Unfortunately, a precipitation series as extensive as Barcelona (1786-present) is unique in our geographical context. On the other hand, for the period 1860-1950 it is possible to carry out these statistical analyses suggested by the reviewer with more data series and with a higher temporal resolution (daily, weekly).

| Index | Pearson correlation | | Spearman correlation | | Kendall correlation | |
|---|---|---|---|---|---|---|
| | Correlation coefficient (R) | Coefficient of determination ($R^2$) | Correlation coefficient (R) | Coefficient of determination ($R^2$) | Correlation coefficient (R) | Coefficient of determination ($R^2$) |
| SPI | -0.62 | 0.38 | -0.39 | 0.15 | -0.28 | 0.08 |
| SPEI | -0.59 | 0.59 | -0.44 | 0.19 | -0.31 | 0.1 |
| Deciles | -0.65 | 0.42 | -0.47 | 0.22 | -0.35 | 0.12 |

Table 7 (Line 746 of the revised manuscript)

Following these results, we have added this text in the discussion section (See Lines 734-740 of the revised manuscript):

*Correlation analyses show moderate to weak negative correlations in all cases. In Pearson's correlation, correlation coefficients range from -0.59 to -0.65, with coefficients of determination ($R^2$) indicating that between 35% and 42% of the variability in the drought indices can be explained by our index. Spearman and Kendall correlations, which do not assume normality of the data, show lower coefficients, suggesting weaker correlations, with $R^2$ values ranging between 0.08 and 0.22. However, given the specific nature and context of our index, it can be considered a suitable proxy for drought, especially when used in combination with other indices and methods of analysis.*

**Bibliography added into the manuscript**

Alcoforado, M. J., Nunes, M. de F., Garcia, J. C., & Taborda, J. P. Temperature and precipitation reconstruction in southern Portugal during the late Maunder Minimum (AD 1675–1715). The Holocene, 10(3), 333–340, https://doi.org/10.1191/095968300674442959, 2000.

Domínguez-Castro, F., Alcoforado, M. J., Bravo-Paredes, N., Fernández-Fernández, M. I., Fragoso, M., Gallego, M. C., García Herrera, R., Garnier, E., Garza-Merodio, G., El Kenawy, A. M., Latorre, B., Noguera, I., Peña-Angulo, D., Reig-Gracia, F., Silva, L. P., Vaquero, J. M., and Vicente Serrano, S. M.: Dating historical droughts from religious ceremonies, the international pro pluvia rogation database. Scientific Data, 8(1), 186, https://doi.org/10.1038/s41597-021-00952-5, 2021.

Espín-Sánchez, J. A., and Gil-Guirado, S.: Praying for rain, resilience, and social stability in Murcia (southeast Spain). Ecology and Society, 27(2), https://doi.org/10.5751/ES-12875-270209, 2022.

Gil-Guirado, S. and Pérez-Morales, A.: Climatic variability and temperature and rainfall patterns in Murcia (1863-2017). Climate analysis techniques in the context of global change. Investigaciones Geograficas, 71, 27–54, https://doi.org/10.14198/INGEO2019.71.02, 2019.

Martín-Vide, J. and Barriendos, M.: The use of rogation ceremony records in climatic reconstruction: a case study from Catalonia (Spain), Climatic Change, 30(2), 201–221, https://doi.org/10.1007/BF01091842, 1995.

Posit team: RStudio: Integrated Development Environment for R. Posit Software, PBC, Boston, MA. http://www.posit.co/, 2024.